# Wind-driven upwelling of iron sustains dense blooms and food webs in the eastern Weddell Gyre

Sebastien Moreau ®[1] ✉, Tore Hattermann[1], Laura de Steur ®[1], Hanna M. Kauko[1], Heidi Ahonen[1], Murat Ardelan[2], Philipp Assmy[1], Melissa Chierici ®[3], Sebastien Descamps ®[1], Tilman Dinter[4], Tone Falkenhaug[5], Agneta Fransson[1], Eirik Grønningsæter[1,6], Elvar H. Hallfredsson ®[3], Oliver Huhn[7], Anais Lebrun ®[8], Andrew Lowther[1], Nico Lübcker ®[9], Pedro Monteiro ®[10], Ilka Peeken ®[4], Alakendra Roychoudhury ®[11], Magdalena Różańska[12], Thomas Ryan-Keogh ®[10], Nicolas Sanchez[2], Asmita Singh ®[10,11], Jan Henrik Simonsen[5], Nadine Steiger[13,14], Sandy J. Thomalla[10,15], Andre van Tonder[16], Jozef M. Wiktor[12] & Harald Steen[1]

The Southern Ocean is a major sink of anthropogenic $CO_2$ and an important foraging area for top trophic level consumers. However, iron limitation sets an upper limit to primary productivity. Here we report on a considerably dense late summer phytoplankton bloom spanning 9000 km$^2$ in the open ocean of the eastern Weddell Gyre. Over its 2.5 months duration, the bloom accumulated up to 20 g C m$^{-2}$ of organic matter, which is unusually high for Southern Ocean open waters. We show that, over 1997–2019, this open ocean bloom was likely driven by anomalies in easterly winds that push sea ice southwards and favor the upwelling of Warm Deep Water enriched in hydrothermal iron and, possibly, other iron sources. This recurring open ocean bloom likely facilitates enhanced carbon export and sustains high standing stocks of Antarctic krill, supporting feeding hot spots for marine birds and baleen whales.

The biological carbon pump in the Southern Ocean, driven by phytoplankton primary production, plays a disproportionately important role in global climate at recent and millennial time-scales[1,2]. This originates from the unique connection between the Southern Ocean and the lower cell of the global overturning circulation[3]. The Weddell Gyre is a key region of the Southern Ocean where Circumpolar Deep Water (CDW) travels through the Antarctic Circumpolar Current (ACC) at mid depth[4], is upwelled to the surface through wind-driven divergence[5]

bringing high concentration of nutrients and carbon to the surface, and modified to Warm Deep Water (WDW) on its path through the Weddell Gyre[6]. Later, it is entrained into the deep ocean as Antarctic Bottom Water (AABW) forming at the southern and southwestern borders of the Weddell Gyre[7]. A recent carbon budget across the Weddell Gyre[8], based on observations of inorganic carbon, suggests that intense primary productivity in the region of the Weddell Gyre, away from continental shelves, can explain the carbon drawdown

[1]Norwegian Polar Institute, Tromsø, Norway. [2]NTNU, Trondheim University, Trondheim, Norway. [3]Institute of Marine Research, Tromsø, Norway. [4]Alfred Wegener Institute, Bremerhaven, Germany. [5]Institute of Marine Research, Flødevigen, Norway. [6]Feltbiologen Grønningsæter, Molde, Norway. [7]Institute of Environmental Physics, University of Bremen, Bremen, Germany. [8]Laboratoire d'Océanographie de Villefranche, Sorbonne Université, CNRS, Villefranche-sur-Mer, France. [9]Department of Biology, University of New Mexico, MSC03-2020, Albuquerque, NM 8713, USA. [10]Southern Ocean Carbon and Climate Observatory, CSIR, Cape Town, South Africa. [11]Department of Earth Sciences, Stellenbosch University, Stellenbosch, South Africa. [12]Institute of Oceanology PAN, Sopot, Poland. [13]Geophysical Institute, University of Bergen and Bjerknes Centre for Climate Research, Bergen, Norway. [14]Sorbonne Université, CNRS/ IRD/MNHN LOCEAN-IPSL, Paris, France. [15]Marine Research Institute, University of Cape Town, Cape Town, South Africa. [16]Mammal Research Institute, Department of Zoology and Entomology, University of Pretoria, Private Bag X20 Hatfield, South Africa. ✉e-mail: sebastien.moreau@npolar.no

estimated from the carbon deficit directed out of the Gyre below the ACC (i.e., at depth between 1500 and 4000 m)[9,10]. The strength of the resulting Weddell Gyre carbon dioxide ($CO_2$) sink to depth, at an estimated ~50 Tg C yr⁻¹, is significant when compared to the global ocean biological carbon pump, estimated at 430 Tg C yr⁻¹ across the 2000 m horizon by Honjo et al.[11]. Satellite-derived ocean color and oceanographic campaigns have however thus far not been able to verify the level of primary productivity required to explain the proposed strength of the Weddell Gyre $CO_2$ sink[8].

Seasonal phytoplankton primary production sustains the Southern Ocean's rich food-web, dominated by the Antarctic krill (*Euphausia superba*), marine birds, seals and both baleen and toothed whales. With over 500 million tons of *E. superba* thriving across its open waters[12], the Southern Ocean ecosystem sustains a rich marine wildlife whose life cycles are linked to the short intensive primary production that takes place in specific areas of the Southern Ocean polar waters.

Here we report such a large scale late-summer phytoplankton bloom as a key feature of the eastern Weddell Gyre ecosystems. The bloom accumulated up to 20 g C m⁻² of organic matter over its 2.5 months duration. A satellite analysis over 1997–2019 shows that this open ocean bloom was likely driven by anomalies in easterly winds that push sea ice southwards and favor the upwelling of Warm Deep Water enriched in hydrothermal iron. This recurring open ocean bloom likely enhances carbon export, sustains high standing stocks of Antarctic krill, and supports feeding hot spots for marine birds and baleen whales.

## Results and discussion
### An anomalously dense open ocean phytoplankton bloom
On 11-12 March 2019, during the *Southern Ocean Ecosystem 2019 cruise*, the RV *Kronprins Haakon* crossed a substantial phytoplankton bloom spanning 9000 km² (300 km long, 30 km wide and aligned along the 3500–4000 m isobath), in the Kong Håkon VII Hav, eastern Weddell Gyre (Fig. 1a). We detected this bloom in an open ocean region with water depths over 3000 m, via analyses of ocean color data derived from ESA and EUMETSAT Sentinel-3A/B satellite images (see Methods). A Seaglider was deployed at the northern edge of the bloom (see Methods for a description of the glider trajectory through the bloom) and observed that the phytoplankton (i.e., chlorophyll *a* (Chl *a*)) and Particulate Organic Carbon (POC), as retrieved from a Chl *a* fluorescence and a backscattering sensor mounted on the Seaglider and calibrated with in situ Chl *a* and POC measurements (see Methods), were distributed from the surface to 50 m depth (to approximately the base of the surface Mixed Layer (ML)). A subsurface Chl *a* and POC maximum was however observed at the base of the ML southwards of 68.32° S (Fig. 1c) where Chl *a* reached 1.9 mg m⁻³ and POC reached 1080 mg m⁻³, which may indicate the beginning of nutrient limitation in the surface mixed layer.

It is believed that we encountered the bloom as it was terminating given that the phytoplankton cells looked chlorotic under light microscopy, the maximum quantum yield of photosystem II, $F_v/F_m$, was lower than the theoretical optimum of 0.65 for phytoplankton growth[13] throughout the bloom area (<0.22, as measured from 5 m depth underway samples, and <0.28 on average at 50 m depth, as measured from samples taken from a CTD-rosette cast at the northern edge of the bloom area, see Methods) and the POC:Chl *a* ratio (g:g) in the ML was elevated[14] (>200, supplementary Fig 1b). In addition, satellite imagery showed that the bloom started in early-January (January 9th), reached its maximum concentration by mid-February (Supplementary Fig 2a), before it collapsed mid-March (March 14th) after our visit of the area[15] (see "Methods" for a description of the MODIS satellite-derived ocean color data and the bloom detection criteria used for this analysis).

This open ocean phytoplankton bloom accumulated striking levels of organic matter over its ~2.5 months duration, with a maximum measured POC concentration of 327 mg m⁻³ at the base of the ML at

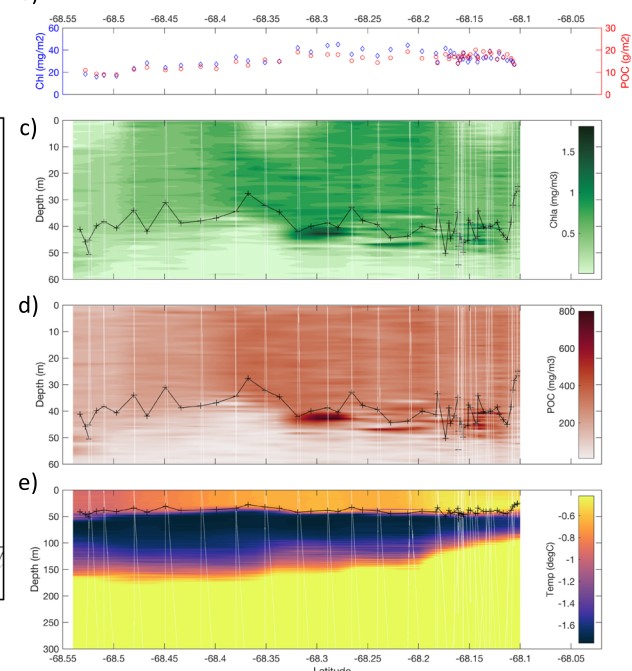

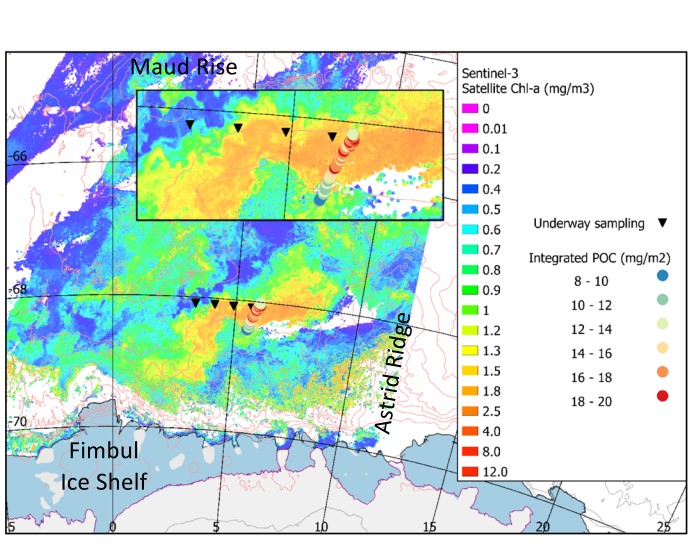

**Fig. 1 | A large phytoplankton bloom in the eastern Weddell Gyre. a** Sentinel satellite image of the open ocean phytoplankton bloom on March 8, 2019. The positions of underwater water samples are indicated where measurements of phytoplankton physiology and microscopic counts and identification of phytoplankton species were carried out (inverted black triangles). Also indicated are the 100 m depth integrated POC (g m⁻²) for each glider profile and the locations of Maud Rise, Astrid Ridge and the Fimbul Ice Shelf. **b** Upper 100 m integrated POC (g m⁻²) and Chlorophyll *a* (mg m⁻²) from each glider profile, **c** contoured Chl *a* (mg m⁻³), **d** contoured POC (mg m⁻³), and **e** contoured temperature (°C) for the glider transect. In panels **c**–**e**, the black line with crosses indicates the depth of the surface Mixed Layer.

the northern edge of the bloom and a maximum estimated POC concentration of 1080 mg m$^{-3}$ at the base of the ML at the core of the bloom (Fig. 1d). Given the high POC:Chl $a$ ratios observed in the ML, it is likely that POC was not only composed of autotrophic but also of heterotrophic carbon. We further estimated that POC had accumulated to 20 g C m$^{-2}$ from the surface to 100 m depth inside the bloom (Fig. 1a, b). Such high POC concentrations and standing stocks in the open Southern Ocean are unusual[16] and comparable to some of the highest concentrations and depth-integrated standing stocks associated with continental shelves and polynyas such as in the Ross and Amundsen seas (maximum POC concentration of 1200 mg m$^{-3}$ and 840 mg m$^{-3}$ and standing stocks up to 108 g C m$^{-2}$ and 38.8 g C m$^{-2}$, respectively)[17,18], naturally iron-fertilized open waters such as in the lee of the Kerguelen Plateau (maximum POC concentration of 384 mg m$^{-3}$ and up to 22.5 g C m$^{-2}$)[19] or artificially fertilized open waters (up to 14.4 g C m$^{-2}$ at the end of the large diatom bloom that was induced by the European Iron Fertilization Experiment (EIFEX) at the Antarctic Polar Front in the Atlantic sector of the Southern Ocean[20]).

To understand the significance and highlight the rarity of such biomass accumulation, we integrated the upper 100 m POC for all the SOCCOM BGC-Argo floats[21] profiles that have been collected in the open Southern Ocean from September 2014 to December 2020, i.e., more than 9500 profiles. We identified only 34 profiles (0.36% of all the SOCCOM BGC-Argo profiles studied) where the integrated POC over the upper 100 m amounted to more than 15 g C m$^{-2}$ and up to 19.7 g C m$^{-2}$ over the whole open Southern Ocean (see "Methods" and Supplementary Fig 3a). These profiles were mainly found in the highly productive Scotia Sea, in the vicinity of Crozet Island and the Kerguelen Plateau, and around Maud Rise close to the bloom we observed in the eastern Weddell Gyre, highlighting the overall importance of the reported bloom in the usually High Nutrient, Low Chlorophyll (HNLC) region of the Southern Ocean.

In the Southern Ocean, primary production is usually thought to be primarily limited by iron (Fe)[22] while the concentrations of macronutrients (nitrate, phosphate and silicic acid) are typically high and non-limiting. During our campaign, light and macronutrients were not found to be limiting throughout the open ocean bloom (as shown in Kauko et al.[15]). Therefore, the production of such striking levels of organic matter in the open waters of the Southern Ocean must have involved an important iron source and a relatively low grazing pressure at the onset of the bloom as grazing can dominate phytoplankton loss in the Southern Ocean[23]. During our campaign, dissolved iron (dFe) was relatively elevated[24], and probably not limiting, at the sea surface throughout the larger study area, ranging from 0.65 ± 0.2 nM at Astrid Ridge, 0.4 ± 0.1 nM at Maud Rise and 0.6 ± 0.04 nM at 2 stations studied south of the bloom we report here (Supplementary Fig 2b). Unfortunately, dissolved iron concentrations were not measured directly inside the open ocean bloom area during the campaign and these concentrations should only be considered representative of the larger study area presented in Kauko et al.[15]. The accumulation of both Chl $a$ and POC at the base of the mixed layer and the very low quantum yield of photosystem II, $F_v/F_m$ may, however, indicate that iron was becoming limiting at the time of sampling.

We calculated the iron necessary to accumulate POC levels up to 20 g C m$^{-2}$ to be between 3.8 and 33.3 μmol Fe m$^{-2}$ by using lower and upper bound Fe:C ratios from Fe-replete and Fe-deplete Southern Ocean waters (2.3 to 20 mol:mol 10$^{-6}$)[25]. Uncertainties on the lower and upper limit of this range are large as they do not account for the role of grazing or vertical export on POC in the ML[23], and as POC is both constituted of autotrophic and heterotrophic carbon. A lower estimate could be obtained by considering that phytoplankton only represents 20 to 30% of the total POC[26]. With this consideration, the necessary iron to accumulate 5 g C m$^{-2}$ (i.e., 25% of 20 g C m$^{-2}$) was calculated to be between 1 and 8.3 μmol Fe m$^{-2}$. An estimation of satellite-derived net primary productivity and the associated iron demands during the 2.5 months bloom period ranged between 14.8 and 42.6 μmol Fe m$^{-2}$ (see Methods). Together, these observations and calculations highlight that an important iron source or mechanism was required in order to lead to such high levels of POC accumulation inside the open ocean bloom.

## A recurring phytoplankton bloom

In order to put the observed bloom in a wider context, we inspected satellite-derived ocean color to detect the presence of the open ocean bloom from September 1997 to December 2019 (see "Methods"). Since the surface area of the bloom as observed from satellite-derived Chl $a$ in March 2019 follows the deep bathymetric contours at 3500 and 4000 m depth (Fig. 1a), we defined the boundaries of this part of the study as the core of the bloom as we observed it in March 2019 and in other years, averaged over 4°–8° E and 67.8°–68.4° S. We defined the bloom presence when the mean satellite-derived Chl $a$ (obtained at 8-day intervals), was larger than the 23-year long mean Chl $a$ + 1 standard deviation over that area (i.e., 1.14 mg m$^{-3}$, Fig. 2a). Results show that the bloom occurred in 9 years out of the 22-year period, mostly in February (55% of occurrences, Supplementary Fig. 4a) but always between January 1 (33% of occurrences were in January) and March 15 (13% of occurrences were in March). The bloom phenology in the area (calculated following Kauko et al.[15]) was not significantly different ($p > 0.05$) between years with and years without an open ocean bloom, and typically initiated on January 15th ±4 days, ended on March 1$^{st}$ ± 3 days, and lasted 44.5 ± 4 days on average.

A composite map of satellite-derived Chl $a$ concentration, obtained by averaging de-seasoned anomalies during time periods when the bloom was present shows a large area of positive Chl $a$ anomaly in the open ocean oriented along a northeast-southwest pathway, extending from 55° S to 30° E to 10° W to 70° S (Fig. 2b and see "Methods" for computation details). The composite map during the bloom presence represents 16% of the total available (monthly de-seasoned) values. Furthermore, a composite map of the de-seasoned zonal (i.e., east-west) wind anomalies during the bloom presence shows a negative anomaly in the area and southwest of the bloom (Fig. 2d) and implies that coastal easterlies are stronger during bloom events (Supplementary Fig 4b). Such wind conditions cause enhanced southward Ekman transport toward the coast thereby enhancing upwelling of iron-rich WDW in the divergence zone[27], north of the band of coastally enhanced prevailing easterly winds in this region. On the contrary, when the open ocean bloom is absent, we observe weak positive anomalies of the zonal wind component, i.e. westerly wind anomalies associated with downwelling anomalies in the open ocean (Fig. 2e). A weak but significant negative correlation exists between the satellite-derived Chl $a$ de-seasoned anomaly and the zonal wind de-seasoned anomaly in the area of the open ocean bloom (with a maximum $r$-value of −0.32; see Methods and Supplementary Fig 5a, b). The correlation is strongest with a 1-month lag (compared to 0 and 2-months' time lag), which is possibly linked to the time needed for a Southern Ocean phytoplankton bloom to develop after iron enrichment (~2-4 weeks[14,20]).

Furthermore, at times when the open ocean bloom takes place, the stronger easterlies push sea ice south, towards the coast. This leads to a negative anomaly in sea-ice concentration in the open ocean west of Astrid Ridge and the opposite positive anomaly in sea-ice concentration along the coast between Astrid Ridge and the Fimbul Ice Shelf Tongue (Fig. 2f). By pushing the sea ice south, towards the coast, enhanced easterlies may also relieve phytoplankton production from possible light limitation by the sea ice cover[28].

The timing of the wind and sea ice conditions are important for the open ocean bloom. In this region, zonal winds typically increase from January onward, peaking in May (Supplementary Fig 4c) while sea ice cover typically reaches its minimum in February-March

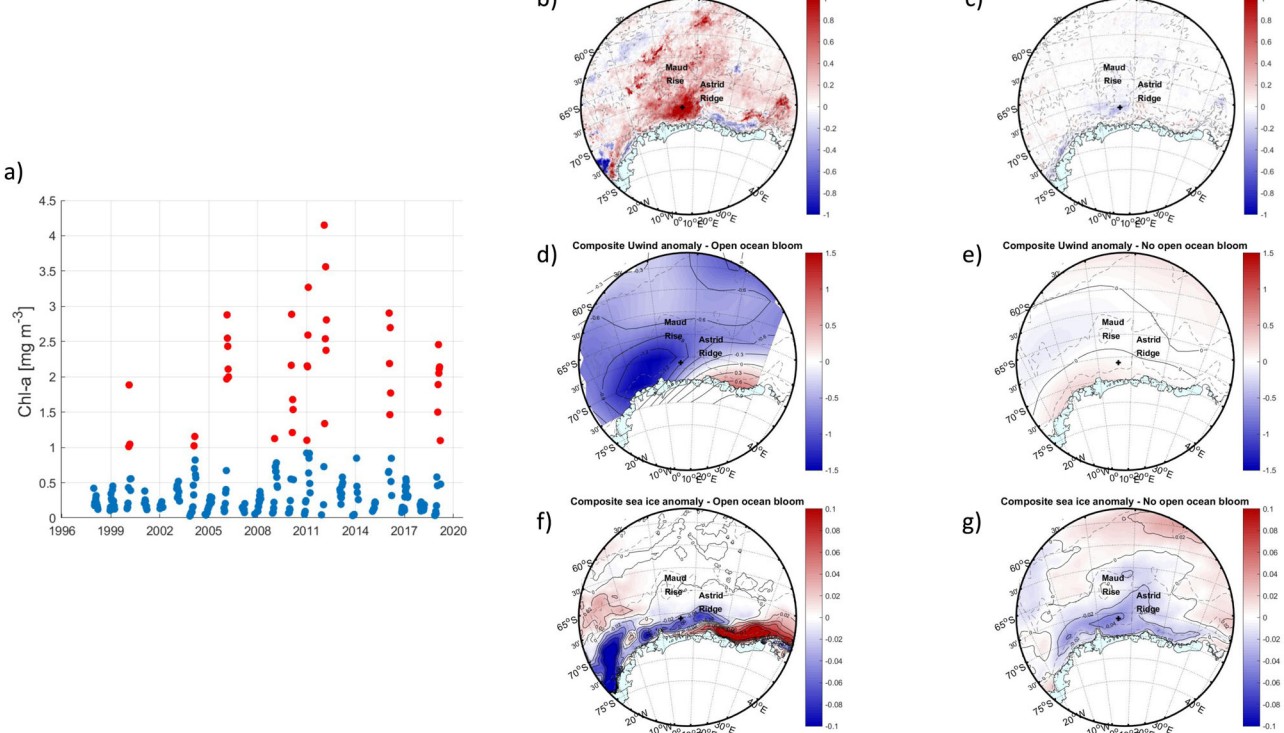

**Fig. 2 | Composite anomalies in the presence and absence of the open ocean bloom. a** Mean satellite-derived Chl $a$ (at 8-day intervals) averaged over 4°–8° E and 67.8°–68.4° S between 1997 and 2019 (all circles). Red circles correspond to when the mean satellite derived Chl $a$ was larger than the 23-year long mean Chl $a \pm 1$ standard deviation averaged over that area (i.e., 1.14 mg m$^{-3}$) and are considered as bloom occurrences. Composite of **b** the Chl $a$ concentration, **d** the zonal wind component and **f** sea-ice concentration de-seasoned anomalies (mean seasonal cycle removed) in the presence of a bloom (the weeks/years indicated by red circles in (**a**)) between 4°–8° E and 67.8°–68.4° S. Composite of **c** the Chl $a$ concentration, **e** the zonal wind component and **g** sea-ice concentration de-seasoned anomalies in the absence of a bloom (the weeks indicated by blue circles in (**a**)) between 4°–8° E and 67.8°–68.4° S. In panels **b**–**g**, the black cross denotes the position of the glider deployment.

(Supplementary Fig 4d). Consequently, the impact of winds on the upper ocean is highest in February-March before sea ice begins to grow. However, when an open ocean bloom occurs, zonal winds peak earlier in the season, between February-April (Supplementary Fig 4c). This induces strong ocean surface stress and provides favorable preconditioning for an upwelling of iron-rich WDW, which likely sustains the large phytoplankton bloom through February-March (Supplementary Fig 4a). This also provides an explanation as to why we typically observe the open ocean bloom in February-March, late in the summer production season.

Entrainment by convective mixing during winter has been highlighted as another mechanism that can make iron from deeper water masses available to the surface with fluxes of up to 33.2 µmol Fe m$^{-2}$ yr$^{-1}$ being observed[29]. Based on existing Argo float profiles, we tested the hypothesis that the bloom magnitude was linked to the preceding winter maximum ML depth from 2007 to 2019 (see "Methods"). We found that no significant differences existed in preceding winter (i.e. September) maximum ML depths between years with (mean of 124.9 ± 6.3 m) and years without (mean of 113.1 ± 13.3 m) an open ocean bloom ($p > 0.05$, Supplementary Fig 4e).

Finally, the bloom seems to be associated with an oceanic front that follows the deep bathymetric contours at 3500 and 4000 m depth (Fig. 1a). A sharp gradient in water temperature was observed across the blooming waters between 100 and 150 m depth, with a deepening of density isopycnals southward that suggests a local frontal structure (Fig. 1e). Mixing of water masses with different origin associated with this frontal structure could help to maintain the bloom by bringing deep iron to the ML under upwelling favorable wind conditions in this region.

**Potential iron sources**

Our regional analysis of more than two decades of satellite-derived Chl $a$ and climatological sea ice and wind data suggests that the observed bloom is associated with anomalous wind-driven upwelling of iron from deeper waters in this region. Such a mechanism assumes that upstream sources provide the necessary iron that is needed to allow for such high biomass build-up. Hydrothermal iron has recently been shown to be transported thousands of kilometers from its origin[30] and be a key driver of primary production of regions that are of critical importance for the oceanic carbon cycle[31]. For example, Ardyna et al.[32] highlighted the possible role of iron from hydrothermal vents located west of the Southwest Indian Ridge (SWIR, yellow star in Fig. 3a) in the unusually large phytoplankton blooms they observed a few hundred kilometers downstream.

The helium isotopic ratio ($\delta^3$He $= 100 \times ((^3$He/$^4$He)observed/($^3$He/$^4$He)atmospheric $- 1$) [%]) is a tracer of primordial helium originating from the Earth's mantle and elevated $\delta^3$He ratios (10–12%) are commonly used to detect plumes downstream of hydrothermal vents[30]. Two sections of $\delta^3$He from the Global Ocean Data Analysis Project (GLODAP-2) across the Weddell Gyre, around 0° and 30° E (see "Methods"), and an additional section of $\delta^3$He from a recent expedition along 6° E (Transekttokt 2020/21, see "Methods") support hydrothermalism as a major iron source for the open ocean bloom we describe here (Fig. 3). The 0° and 30° E sections show patches of high $\delta^3$He ratios (10–12%) between 500 and 100 m depth in the northern part of the section around 50–55° S (Fig. 3b, d), attributed to deep ocean sources in the ACC region. At 30° E, these patches extend all the way to 60° S (Fig. 3d), likely indicating the eastward advection of $\delta^3$He enhanced sub-surface water masses originating from the

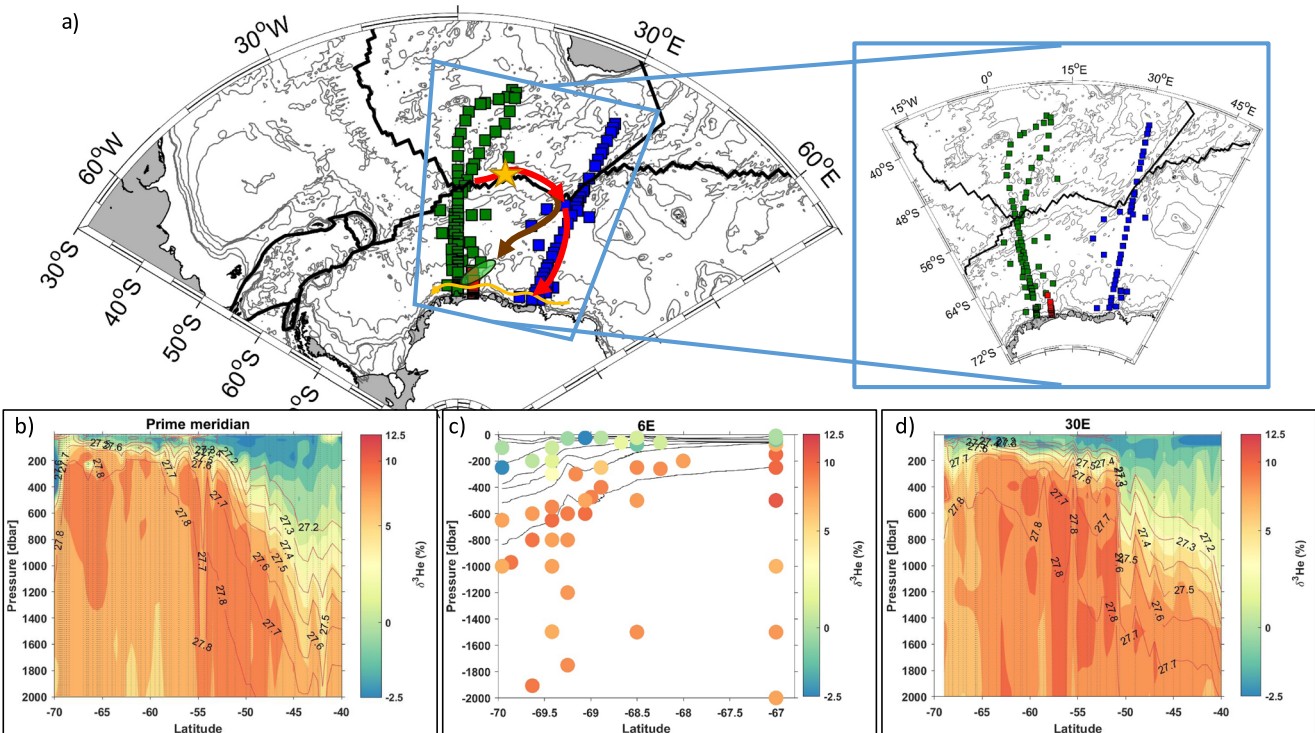

**Fig. 3 | Hydrothermal influence in the Eastern Weddell Gyre. a** Map of the locations of isotopic helium ($\delta^3$He) data in the water column from the GLODAP database[41] and the Transektokt 2020/21, at or close to the Prime Meridian (green squares), following a transect along 6° East (red squares) and a transect along 30° East (blue squares). Also indicated by colored lines and arrows are the tectonics plates boundaries (black line) and a sketched circulation of the eastern Weddell Gyre from Ryan et al.[33] who described a two-core pathway of the northeastern inflow of WDW at about 20° E, the northern pathway being driven by eddy mixing in the northeastern corner of the Weddell Gyre (brown arrow), the southern one being

an advective route which forms the southern branch of the inflow and extends beyond 30° E before turning westward (red arrow). The yellow arrow indicates the predominantly westward coastal flow regime south of 65° S[35]. The presence of the bloom is indicated by a green oval. The yellow star indicates the presence of hydrothermal vents at the Southwest Indian Ridge[36]. Contoured and scatter plots of $\delta^3$He (%) at or close to the Prime Meridian (**b**), following a transect along 6° East (**c**), and along 30° E (**d**). Density isopycnal contours are also indicated. Note the different latitude range used for the 6° East transect.

hydrothermal vents located west of the SWIR (red arrow in Fig. 3a). However, in the predominantly westward coastal flow regime south of 65° S (yellow arrow in Fig. 3a), sub-surface $\delta^3$He ratios are higher at the Prime Meridian than at 30° E (Fig. 3b), indicating an additional source of hydrothermally enriched waters between these two sections that are located upstream and downstream of the bloom region (green patch in Fig. 3a). We suggest that this additional source is associated with the northeastern inflow pathway of the Weddell Gyre[33], which brings hydrothermally enriched waters from downstream of the SWIR southward towards the bloom region (brown arrow in Fig. 3a), where iron is made available for biological consumption by local upwelling. This is supported by the $\delta^3$He section along 6°E where elevated $\delta^3$He ratios are found close to and within the surface mixed layer at the northern end of the transect (9.8 and 6.9% at 150 and 75 m depth, respectively, Fig. 3c). In this way, hydrothermal vents may act as the long-range source of iron that sustains and explains the persistence of the open ocean bloom that we observed.

While the above analysis points at hydrothermalism as an important source of iron for the observed bloom, microbial iron remineralization, which can provide 5–10 μmol m$^{-2}$ d$^{-1}$ of dissolved iron[34], may have subsequently supplied a substantial portion of the iron required to maintain the high biomass bloom. As highlighted above, entrainment of WDW into the surface layer by convective mixing during winter is an important source of iron before the start of the productive season in the Southern Ocean (up to 33.2 μmol Fe m$^{-2}$ yr$^{-1}$)[29], even though entrainment anomalies are not linked to bloom anomalies in the present study (Supplementary Fig 4e). Furthermore, the bloom was found over deep waters (3000 m depth), far from the

coast (200 km), and not in the lee of any shallow continental shelf or plateau. However, similar to hydrothermal iron, sedimentary iron[19] may have been transported over long distances beneath the mixed layer and brought up to the surface layer by the upwelling mechanism described above.

Considering other known iron sources, atmospheric deposition of iron plays a minor role at these latitudes[35]. The bloom was far from typical iceberg drifting routes, with no signs of glacial meltwater in water mass characteristics. In addition, the contribution of glacial meltwater is of minor importance because of the generally low melt rates of the ice shelves in this region[36], ruling out the likelihood of glacial-derived iron[37]. Moreover, we detected no signs of sea-ice meltwater in the water column derived from stable oxygen isotopic ratios ($\delta^{18}$O) (between −0.07 at the surface and 0.4‰ at 1000 m depth) from samples collected by a CTD-rosette cast at the northern edge of the bloom (see Methods), suggesting that sea ice meltwater is similarly not a major source of iron for this open ocean bloom[38].

**Phytoplankton community composition and potential for carbon export**

In addition to high biomass, the taxonomical composition of this open ocean bloom has important biogeochemical and ecological consequences. The bloom was dominated by the centric diatom *Chaetoceros dichaeta*, which represented 39–67% of the total phytoplankton abundance as evidenced from surface underway samples (intake at 4 m depth) and between 31 and 47% of the total phytoplankton abundance within the upper mixed layer, i.e., down to 40 m depth as evidenced from CTD stations (locations of sea surface underway and

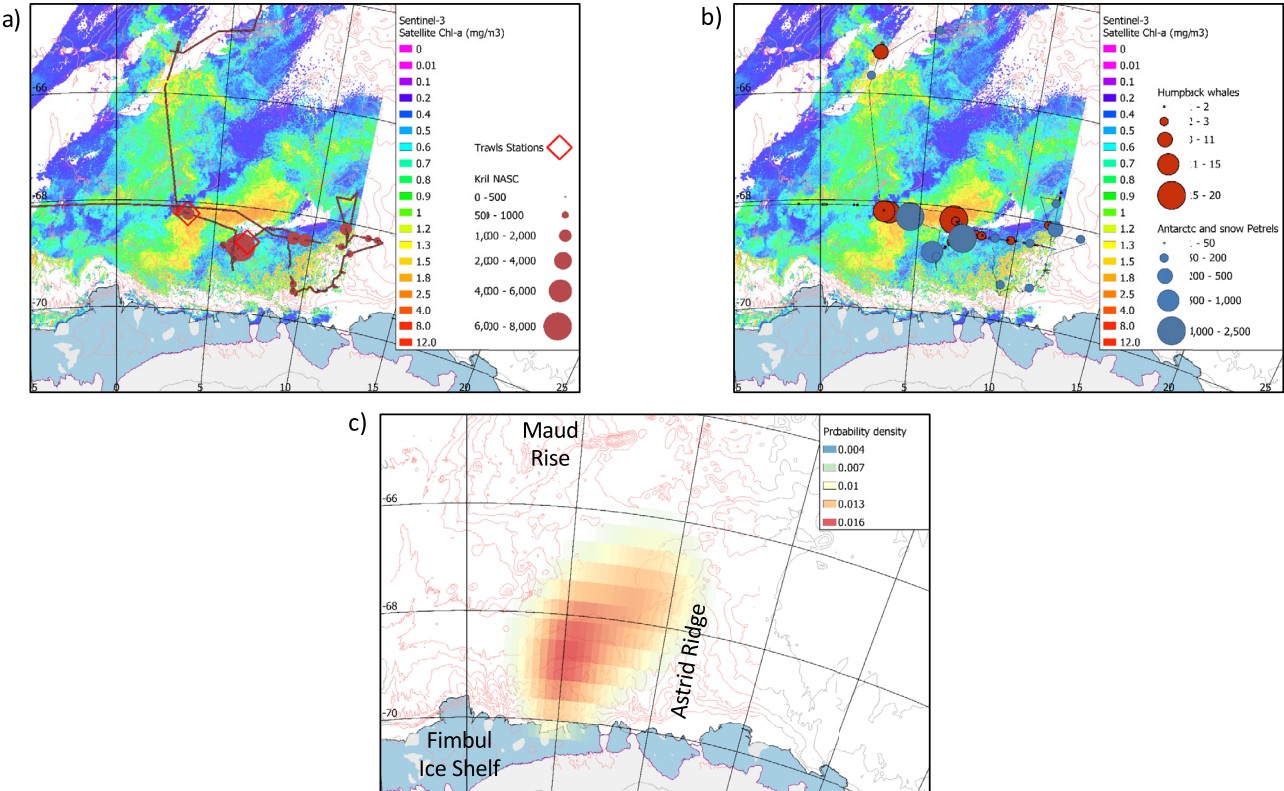

**Fig. 4 | The open ocean bloom attracted a rich ecosystem in the open Southern Ocean. a** Abundance of krill (nautical area scattering coefficient (NASC) values integrated over 500 m distance) retrieved from the ship's echosounders during the ecosystem cruise. Also indicated are the two krill trawling stations inside the bloom area (red diamonds). **b** Abundance of humpback whales (number of individuals per sighting event) and Antarctic and Snow petrels (total daily sightings) observed during the ecosystem cruise. **c** Heat map of the utilization distribution, (i.e., the probability density that an animal is found at a point according to its geographical coordinates) of Svarthamaren Antarctic petrels averaged over the summers 2012, 2013, 2014, 2016 and 2018 (see "Methods"). In panels **a** and **b**, the Sentinel satellite image of the open ocean phytoplankton bloom on March 8th, 2019 is used.

water column sampling are provided in Fig. 1a). At the northern edge of the bloom area, where we deployed the Seaglider, the abundance of *C. dichaeta* was $1.5 \times 10^6$ cells $L^{-1}$, which is very large in comparison to the total diatom and *C. dichaeta* abundances of $0.5 \times 10^6$ cells $L^{-1}$ and $0.03–0.055 \times 10^6$ cells $L^{-1}$, respectively, during the EIFEX fertilization experiment (P. Assmy, pers. comm.). The second most abundant phytoplankton species in this blooming assemblage was the centric diatom *Dactyliosolen antarcticus*, but it only accounted for a maximum of 10% of total phytoplankton abundance on any occasion.

*Chaeotoceros dichaeta* is a large (~10 to 50 μm) chain-forming diatom species with prominent spines. It can dominate phytoplankton blooms in the coastal and the sea ice zone around Antarctica, particularly in summer, whereas, in frontal regions, it can be dominant in the fall[39]. Due to their prominent spines, members of the *Chaetoceros* genus, including *C. dichaeta*, easily form rapidly sinking aggregates when undergoing mass mortality and are considered to be important carbon exporters[40]. The large bloom of *C. dichaeta* we observed over deep waters in the eastern Weddell Gyre was actively contributing to deep carbon export as shown by the large optical spikes created by particles and phytodetritus (i.e Chl *a*) below 100 m and throughout the bloom area (Supplementary Fig 1c, d). These optical spikes only provide an instantaneous view of the contribution of the bloom to deep export as we lack a Lagrangian time series inside the bloom to properly assess biological carbon export[23]. In addition, on board measurements of sea surface fugacity of $CO_2$ indicate that the bloom region was a net sink for atmospheric $CO_2$[41]. Although it remains uncertain whether this carbon ultimately reached the seafloor, large pulsed bloom events, such as the bloom of *C. dichaeta* we report here, might by-pass the shallow

remineralization horizon (100–250 m deep)[42] and efficiently export carbon to the deep sea and the sea floor. A similar event could be shown during the EIFEX fertilization experiment which took place north of the ACC, northeast of the Weddell Gyre, where *C. dichaeta*, contributed significantly to carbon export 24 days after bloom fertilization[20]. In this instance, half of the bloom biomass sank below 1000 m, and a phytodetritus layer composed of intact cells of *C. dichaeta* was observed on the seafloor at > 3000 m depth, suggesting that blooms of *C. dichaeta* contribute significantly to deep carbon sequestration in the Southern Ocean.

### Importance for the food-web of the Southern Ocean

In addition to the disproportionately important role of the Southern Ocean's biological carbon pump in global climate, phytoplankton blooms also sustain the Southern Ocean's rich food-web, dominated by Antarctic krill, marine birds, seals and both baleen and toothed whales. During our entire survey across the Weddell Gyre, Antarctic krill (*Euphausia superba*) was mostly observed (with trawl and echo sounding, see Methods) in two areas: in the vicinity of the open ocean bloom we describe here (Fig. 4a), and in Drake Passage. Our survey was conducted relatively late in the season for krill spawning activity, and the krill population was probably in a post-spawning state. This is supported by the maturity stage composition observed in two krill trawl catches within the bloom region, which happened during our second passage inside the bloom region (24–27 March, Fig. 4a) after the bloom had decayed. These catches were dominated by adults (54.2 and 58.4%) and contained very few juveniles (3.9 and 6.5%) (see Supplementary Table 1). The observation of few adult females with swollen and spent ovaries also supports this interpretation.

The krill observed in the open ocean bloom area had green colored digestive glands and stomachs, indicating a phytoplankton based diet. From water column samples, we also observed high ratios of the grazing pigment pheophorbide *a* to Chl *a* of almost 0.5 that indicated a high degree of grazing only observed inside the bloom[43]. Therefore, it is likely that Antarctic krill was actively grazing on the large and high-biomass phytoplankton bloom we report here. The Kong Håkon VII Hav, in the eastern Weddell Gyre, has been described as one of the three major krill development areas[44], along with the Scotia Sea and Antarctic Peninsula; and the area north of Prydz Bay and the Kerguelen Plateau. In addition, there exist two main distributional centers, or areas of highest densities, of Antarctic krill, one aligns the ACC stream in the Scotia Arc and the other lies along the shelves of the Kong Håkon VII Hav and Cosmonaut Sea[12]. Furthermore, it has been hypothesized that the Kong Håkon VII Hav is a source region of larval and juvenile krill for the larger Scotia Sea krill stock[45], and Siegel[46] recognized that the Kong Håkon VII Hav is an active and successful spawning ground for Antarctic krill. This evidence suggests that the persistently occurring bloom we describe here helps to sustain the Antarctic krill population in this key region.

In addition to Antarctic krill, we observed very high abundances and densities of Antarctic and snow petrels (*Thalassoica antarctica* and *Pagodroma nivea*, Fig. 4b) in the open ocean bloom area. Maximum abundances were, respectively, 1480 Antarctic petrels and 805 snow petrels in two separate days of observations (24 and 27 March 2019), which corresponded to 263 Antarctic petrels/km$^2$ and 91 snow petrels/km$^2$. This represents some of the highest abundances and densities reported at sea in this part of the Weddell Gyre and the Southern Ocean at large[47]. For comparison, we observed low abundances of flying seabirds (average of $19 \pm 3$ daily sightings, all species combined) outside the open ocean bloom and throughout our entire survey across the Weddell Gyre. In addition, we observed high numbers of humpback whales foraging in the area of the open ocean bloom (i.e. 62 and 78 humpback whales in two separate days, 12 and 27 March 2019, Fig. 4b), for a total of 228 humpback whales sighted during the entire campaign (41 days and a total sailing distance of 5595 Nm) and a daily average sighting of $2.4 \pm 0.6$ humpback whales outside the bloom. In comparison, during the entire Broke-West campaign in East Antarctica, Nicol et al.[48] observed 150 humpback whales in 77 sightings over 2 months (February–March 1996), but did not report similarly high local abundance numbers like the ones we observed in the bloom area. As shown in previous studies[48], the distribution of flying seabirds and baleen whales reflects the predictable distribution of their main prey, the Antarctic krill. Krill is a key prey item for numerous Antarctic species because of the extraordinary size of their swarms[49] and their overall dominance in the Southern Ocean[12], making it an easily accessible and profitable prey. It is, therefore, very likely that the large number of flying seabirds and baleen whales we observed in the open ocean bloom area were attracted by the presence of Antarctic krill.

Dronning Maud Land hosts a large population of seabirds, and in particular half of the World's Antarctic petrel population, which represents four to seven million breeding Antarctic petrels[50]. Using four years of tracking data, Fauchald et al.[51] suggested that, in austral summer, Antarctic petrel breeding at Svarthamaren, 200 km south inland of the open ocean bloom presented here, forage mostly in areas that have reached an age of 50 to 60 days from the date of sea ice melt. The authors suggested that this time lag corresponds to the necessary time for a short-lived phytoplankton bloom to develop following the sea ice melt and be subsequently exploited by a profitable life-cycle stage of Antarctic krill, i.e. gravid and spawning krill. Distribution maps of foraging Antarctic petrels during the austral summer show that Antarctic petrels spend most of their time at sea directly at or in the direct vicinity of the open ocean bloom (Fig. 4c and Supplementary Fig 6; see also Fig. 1 in Descamps et al.[52]). This possibly brings new insight into the foraging strategy of Antarctic petrels along the

Dronning Maud Land coast and highlights this open ocean bloom as their possible favored feeding ground during the breeding season.

Our study provides further evidence that humpback whales feed in the open waters of the eastern Weddell Gyre. These findings are consistent with two recent acoustic studies showing that humpback whales utilize this area during the typical open ocean bloom period: humpback whale vocalizations were recorded from March to May at Maud Rise and almost throughout the year near Ekström Ice Shelf with peak occurrence in February to May[53,54]. Therefore, our observations along with the results from previous studies highlight this region as an important foraging area for Antarctic wildlife. Although unpredictable, these re-occurring open ocean blooms are a possible key feature of the Weddell Gyre marine ecosystem.

## Perspectives

The evidence we report here points towards the importance of this open ocean bloom for the eastern Weddell Gyre ecosystem; particularly the characteristics directly observed at sea (i.e. high phytoplankton carbon biomass and abundant krill), its recurrence during favorable atmospheric and oceanographic conditions (i.e., stronger easterlies, upwelling and less sea ice), the likely additional sources of iron sustaining it (including hydrothermal iron, entrainment and remineralization), and its seemingly long-term significance for the biological carbon pump and top trophic level consumers (marine birds and whales). The timing of the wind and sea ice conditions are important for the open ocean bloom which typically happens in late summer, i.e., most often in February. Such a productive bloom late in the summer is significant for example for krill that need to prepare for the long austral winter[55] or for Antarctic petrels that are raising and fledging chicks in February–March[51].

The next legitimate question concerns the fate of this open ocean bloom and its associated ecosystem in the context of climate change. A recent study described major changes in the Southern Ocean physical and biogeochemical properties[56] due to poleward intensifying winds and increases in meltwater over the last two decades, which could have large consequences on the fate of this open ocean bloom. Given its seemingly importance for the Weddell Sea, the driving forces and the fate of this key ecosystem feature warrant future studies.

## Methods
### Seawater collection and sample analysis
Seawater was collected from the ship's scientific seawater intake at 4 m depth throughout the large phytoplankton bloom ($N = 5$) as well as from a SBE 32 carrousel water sampler connected to a CTD (conductivity-temperature-depth) SBE911+ system which was deployed to calibrate the glider sensors at the glider deployment station (9 depths: 150, 125, 100, 75, 50, 40, 25, 10 and 5 m). Seawater was sampled to measure the concentrations of chlorophyll *a* (Chl *a*, measured by High Performance Liquid Chromatography (HPLC), Particulate Organic Carbon and Nitrogen (POC and PON), the phytoplankton species composition, phytoplankton physiology, and stable oxygen isotopic ratio ($\delta^{18}$O)).

Water samples (1 L) for HPLC analyses were filtered on 25-mm Whatman 0.7 um GF/F glass fiber filters (GE Healthcare, Little Chalfton, UK) under low vacuum pressure and then stored at $-80\,^{\circ}$C until analysis ashore. Chlorophyll *a* was measured with reverse-phase HPLC including a Agilent Technologies Microsorb-MV3 C8 column ($4.6 \times 100$ mm), a Waters Alliance HPLC System and a photodiode array detector (2996). HPLC-grade solvents (Merck) and the EMPOWER software were used in the analysis. More details can be found in Tran et al.[57].

Water samples (1 L) for POC analyses were filtered on precombusted 25-mm Whatman 0.7 um GF/F glass fiber filters (GE Healthcare, Little Chalfont, UK) under low vacuum pressure, dried at $60\,^{\circ}$C and stored at room temperature in PALL filter slides until

analyses ashore. 80 µL of 10% v/v hydrochloric acid was added to each filter to remove any inorganic carbon before the filters were folded into aluminum capsules. POC concentrations were then measured with a C/N analyzer Europa Scientific, ANCA-MS 20-20 15N/13C mass spectrometer at the Tvarminne Zoologica Station, Finland. POC values were corrected for dry filter blanks.

Samples for phytoplankton species counts by microscopy (190 mL) were filled into 200 mL brown glass bottles and fixed with 25% glutaraldehyde and 20% hexamethylenetetramine-buffered formalin solution at final concentrations of 0.1 and 1%, respectively, and thereafter stored cool and dark until analyses ashore. For analysis, 10–50 mL Utermöhl sedimentation chambers (HYDRO-BIOS©, Kiel, Germany) were used for settling sub-samples for 48 h and Nikon Ti-S inverted light microscope for identification and counting by the Utermöhl method[58].

Measurements of phytoplankton photophysiology were performed using a Chelsea Scientific Instruments FastOcean™ fast repetition rate fluorometer (FRRf) integrated with a FastAct™ laboratory system. After collection, samples were dark acclimated at in situ temperatures for 30 min before measurement. The active Chl $a$ measurements consisted of a single turnover protocol with a saturation sequence ($100 \times 1$ µs flashlets with a 2 µs interval) and a relaxation sequence ($25 \times 1$ µs with an interval of 84 µs). The sequence interval was 100 ms which was repeated 32 times resulting in a total acquisition time of 3.2 s. The power of the excitation LED ($\lambda 450$ nm) was adjusted between samples to saturate the observed transients following the manufacturer specifications. Each sample underwent a fluorescence light curve which measured the active Chl $a$ fluorescence at 15 light levels between 0–1500 µE m$^{-2}$ s$^{-1}$, with an optimized duration per light level consisting of 12 acquisitions per light level, except for the first light level (10 µE m$^{-2}$ s$^{-1}$) which consisted of double the number of acquisitions (24). All samples were blank corrected using carefully prepared 0.2 µm filtrates[59].

For $\delta^{18}O$ analyses, samples were collected in 25 mL Wheaton bottles. The caps were sealed with Parafilm® and the bottles were stored in +4 °C and dark conditions until analysis in the laboratory. The analysis of $\delta^{18}O$ was done with a Thermo Fisher Scientific Delta V Advantage mass spectrometer with Gasbench II. The $\delta^{18}O$ data were standardized to Vienna Standard Mean Ocean Water (VSMOW). The reproducibility of the replicate analyses was ±0.04‰.

For the analyses of Dissolved labile (dFe) and total acid leachable (TaLFe) iron, water samples were collected at 12 stations with 5 L acid-cleaned Teflon-lined GO-FLO bottles deployed on a 500 m aramid rope (6 mm-vpg industri), using a dedicated winch and Teflon coated messengers. The samples were collected at 8 discrete depths (20, 30, 50, 75, 100, 200, 300 and 500 m). After water collection, the GO-FLO bottles were double bagged and taken to a makeshift clean room, placed on a rack under clean air blowing from a Class-100 laminar flow hood and drained into acid washed 125 mL low-density polyethylene (LDPE) Nalgene bottles. Water for TaLFe determination was collected directly into the LDPE without filtration. For dFe samples, water was filtered through Sartorius filters (0.45 + 0.2 µm pore size filtration) using acid-washed Tygon tubes. The dFe fractions were defined operationally by the 0.2 µm nominal pore size. During filtration, an additional HEPA air-filter cartridge (HEPA-CAP/HEPA VENT, 75 mm, Whatman) was connected to the pressure relief air-vent valve of the GO-FLO bottles to ensure that the air in contact with the sample during the filtration was clean. All samples for dFe and TaLFe were acidified to pH 1.7–1.8 with ~3 M double quartz distilled HNO3 (UP HNO3). The acidified water samples were stored (> 1 year) until analysis at Stellenbosch University (TracEx, South Africa) as described in Samanta et al.[60] using online pre-concentration methods. All samples were measured in duplicate. The detection limit of Fe was 0.08 nM.

## Seaglider deployment and data treatment

An autonomous Seaglider (SG563) was deployed and recovered during the cruise, collecting vertical profiles during 31 dives along a 55 km transect across the bloom region between March 12 and March 15, 2019 (Fig. 1a). The glider was remotely operated by Norgliders (http://norgliders.gfi.uib.no/), profiling continuously between the surface and 1000 m depth and measuring a suite of parameters that includes conductivity (salinity), temperature, depth (CTD), dissolved oxygen, Chl $a$, and optical backscattering by particles as a proxy for POC. The Seaglider Chl $a$ fluorescence and backscattering signals were calibrated against the Chl $a$ and POC concentrations measured during a calibration CTD cast shortly after the Seaglider deployment (2.5 h). The Seaglider Chl $a$ fluorescence and backscattering signals were processed following the glider toolbox described by Gregor et al.[61]. In situ dark counts were calculated from the 95th percentile between 400 and 600 m and removed from the data. A 3-point rolling mean was subtracted from each profile to de-spike fluorescence and backscattering profiles. The mixed layer depth (ML) for each profile was computed as the depth to which temperature exceeded 0.04 °C of the surface temperature, which was a good indicator of the bottom of the ML based on visual inspection of the profiles and general Temperature and Salinity properties in the area[15].

## Satellite data (Chl a, sea ice and winds)

The Chl $a$ concentration maps are extracted from the Ocean and Land Colour Instrument (OLCI) Level-2 full resolution Near Real Time (NRT) product, obtained by the EUMETCast broadcast system. The OLCI sensors on board ESA Sentinel-3A and B satellites launched in February 2016 respectively April 2018, and have large swath widths (-1270 km) covering large regions with high temporal resolution (approximately 1.5-day global coverage with both sensors). The OC4ME algorithm which uses a polynomial approach of a maximum band ratio algorithm of 4 reflectances at 443, 490 and 510 nm over the 560 nm was extracted directly from OLCI Level 2 products and gives the Chl $a$ pigment concentration in [mg/m3]. The cloud free chlorophyll orbit tiles are binned and averaged on a daily base to obtain one image per day. This leads to a very good resolution, coverage in reasonable timeframe and meets the requirements of monitoring water dynamics also on smaller spatial scales.

The analysis of satellite-derived ocean color was complemented using SeaWiFS (1997–2002, https://oceandata.sci.gsfc.nasa.gov/SeaWiFS/) and MODIS (2003–2020, https://oceandata.sci.gsfc.nasa.gov/MODIS-Aqua/) level 3 Chl $a$ data processed with the default chlorophyll algorithm (chlor_a) which employs the standard OC3/OC4 (OCx) band ratio algorithm merged with the color index (CI) in 8-day and 9 km resolution.

First, the long-term analysis of the bloom presence (Fig. 2a) was performed from January 1997 to December 2020, covering the ocean-color satellite era. We defined the bloom presence when the 8-day mean satellite-derived Chl $a$, averaged over 4°–8° E and 67.8°–68.4° S, was larger than the 23-year long mean Chl $a$ + 1 standard deviation over that area (i.e., 1.14 mg m$^{-3}$, Fig. 2a).

In addition, the bloom duration in 2019 was calculated as the period between the first occurrence of the bloom during the Austral summer 2019 and its end, before the following Austral winter. The long-term median of the average Chl $a$ concentration over the area 4°–8° E and 67.8°–68.4° S was used as a threshold to detect the bloom onset and end above which we considered a phytoplankton bloom present. Because of the presence of clouds and sea ice, satellite-derived ocean color did not detect any pixel with Chl-a data in the bloom area before January 9, 2019, and after March 14, 2019, which we, thus, defined as the bloom duration. It is possible, however, that the bloom started earlier and terminated later than our conservative estimate.

Daily sea ice concentration (1997–2020) were obtained from the NASA's Nimbus-7 Scanning Multichannel Microwave Radiometer (SMMR) and Defense Meteorological Satellite Program (DMSP)-F13, -F17, and -F18 Special Sensor Microwave/Imager (SSM/I). Data with a spatial resolution of 25 km were provided by the National Snow and Ice Data Centre, University of Colorado in Boulder, CO (http://nsidc.org), with prior processing using the NASA team algorithms[62].

Daily east-west and north-south wind components at the sea surface were obtained for the study region from the ERA-Interim global atmospheric reanalysis[63].

### De-seasoned anomalies, composite and correlation maps

We computed de-seasoned anomalies of monthly Chl *a*, east-west winds and sea ice concentration by first removing the monthly climatology from monthly means, producing de-seasoned anomalies from September 1997 to December 2019. Thereafter, we produced composite maps by averaging monthly de-seasoned anomalies during and outside of the bloom period. Composites maps of the de-seasoned anomalies are presented in Fig. 2.

Furthermore, correlation maps between the satellite-derived Chl *a* and the east-west wind de-seasoned anomalies and between the satellite-derived Chl *a* and the sea ice concentration de-seasoned anomalies were produced by using the Pearson Correlation test. Correlation maps were assessed for multiple lag periods with the highest correlation coefficients found for winds leading bloom by 1-month.

### Calculation of net primary productivity and iron demands

To calculate Net Primary Productivity (NPP) in the area where we detected the bloom, we used a 2.25-degree bounding box around the position of the station where we deployed the glider and calculated the mean monthly NPP from satellite data products (VGPM, Eppley-VGM, Platt and CbPM). We further converted NPP from all methods into Fe demand ($\mu mol\ m^{-2}\ d^{-1}$) using the mean ratio of Fe:C (4.3 μmol:mol) generated for the Sub-Antarctic Zone of the Southern Atlantic Ocean[64].

### Helium data from GLODAP

The GLODAPv2.2019 dataset[65] was used to retrieve helium data from our region of interest in the Atlantic sector of the Southern Ocean. The GLODAP Version 2 data product was obtained from https://www.glodap.info/.

### Helium data from a recent expedition (Transektokt 2020/21)

Onboard the cargo vessel Malik Arctica in December 2020 to January 2021, seawater was collected from a SBE 32 carrousel water sampler connected to a CTD (conductivity-temperature-depth) SBE911+ system. For helium analyses, water samples were collected into copper tubes, which were clamped off after sampling and later analyzed at the IUP Bremen noble gas mass spectrometry lab HELIS. Samples were processed initially with an ultra-high vacuum gas extraction system. For analysis of the noble gas isotopes [3He, 4He] the extracted gas is transferred into a fully automated mass spectrometric system equipped with a two-stage cryogenic trap system. The system is calibrated using atmospheric air standards on a regular basis and measurements are made for blanks and linearity. For details, see Sültenfuß et al.[66]. The reproducibility is better than ±0.2% and the accuracy better than ±0.5%.

### SOCCOM data

We used all SOCCOM BGC-Argo floats profiles that measured upper ocean POC concentration in the Southern Ocean[21] from September 2014 to December 2020. Data were collected and made freely available by the Southern Ocean Carbon and Climate Observations and Modeling (SOCCOM) Project funded by the National Science Foundation, Division of Polar Programs (NSF PLR −1425989 and OPP-1936222), supplemented by NASA, and by the International Argo Program and

the NOAA programs that contribute to it (https://argo.ucsd.edu/, https://www.ocean-ops.org/board?t=argo). The Argo Program is part of the Global Ocean Observing System. We considered the profiles where the integrated POC in the upper 100 m was above $15\ g\ C\ m^{-2}$. In addition, to avoid comparing our data with episodes of eddy-subduction pump which can send high concentration of organic matter in one short-time process, we removed 15 profiles where the integrated POC in the upper 100 m was above $15\ g\ C\ m^{-2}$ but more than 90% of the POC was found below the surface mixed layer.

### Determination of mixed layer depth in the area

We calculated the mixed layer depth in the area of the bloom between 2002 and 2020 by using all Argo floats that drifted through the area within 4 to 8E and −69 to −67S, i.e., total of 1024 profiles of temperature, salinity and depth. Furthermore, we removed profiles earlier than 2007 due to large seasonal gaps. From all profiles, we calculated the mixed layer depth as the depth where the potential density exceeds its 10 m value by $0.03\ km/m^{3}$ (de Boyer Montegut et al.[67]). We then averaged the obtained mixed layer depths to monthly means.

### Krill detection and collection

The abundance of krill was studied with a Simrad EK80 research echosounder with six frequencies. For this study, we focused on the 38 kHz frequency. It was scrutinized with the Large Scale Survey System (LSSS) software version 2.5.0[68]. During the campaign, we used two transducers: one on the drop-keel (3 m from the hull when down) and one hull-mounted. The latter was used in ice-covered areas, with low impact on the detection of krill swarms[15]. The density of krill was calculated as nautical area scattering coefficient (NASC)[69].

Krill swarms were sampled with a Macroplankton trawl at two locations within the bloom region (the positions of the two trawling stations are indicated in Fig. 4a). The trawl is a fine-meshed plankton trawl with a 36 $m^2$ mouth-opening and 3 × 3 mm diamond shaped mesh (7 mm stretched) from mouth to rear. Towing speed was normally 2.5–3 knots. The velocity of the trawl through water and depth of the trawl were monitored by a depth sensor (SCANMAR) attached to the headline. From each trawl, a subsample of approximately 150 individuals of *Euphausia superba* was taken, and the length of the individual krill was measured (±1 mm) from the anterior margin of the eye to tip of telson excluding the setae. Sex and maturity stages of *E. superba* were determined using the classification methods according to Makarov and Denis[70].

### At-sea observations of marine birds and mammals

At sea observations were done from the ship's 11th deck (21 m above sea level), during daytime and decent weather conditions (i.e., cetacean survey when visibility > 1000 m and wave heights <5 m while the bird survey only required visibility > 300 m). Observations were only recorded during ships transits. Marine mammals were primarily recorded from 270-0° (0° being the bow of the ship), on the port side of the ship. Sea birds were both recorded continuously, counting every bird within 300 m in a sector from the bow (0°) of the ship to 90° in 10 min periods, and with point counts every 30 min, identifying ship's followers behind the ship. As many birds follow the ship for some time, this means double counting the same individuals happens as birds that are counted during a 10-minute period will be counted again during the next 10-min period if they are still present. This method is therefore not for population estimates, but more suitable for density assessment.

### Tracking of Antarctic petrels

Breeding adults were captured on theirs nest during late incubation and mid chick rearing (from early January to mid-February) and instrumented with Global Positioning System (GPS) loggers (CatTrack 1, Catnip Technologies Ltd, Anderson, USA) just before leaving on a

foraging trip. The GPS units weighed approx. 20 g (ca. 3% of bird body mass) and were taped to tail feathers. Birds were recaptured upon return to their nest to retrieve the GPS units and download the data. GPSs recorded the locations of the birds along their foraging trip at 5 to 30 min intervals. In total, data were available from 161 tracks collected in January/February 2012, 2013, 2014, 2016 and 2018. Additional details about the procedure are given in[51,52]. The utilization distribution maps (or heat maps) were created with the kernelUD function of package adehabitatHR in R software version 4.0.2, using the href (reference bandwidth) smoothing parameter (https://www.rdocumentation.org/packages/adehabitatHR/versions/0.4.19/topics/kernelUD).

### Data analysis

Data analyses were conducted with Matlab R2019b. All maps were built on QGIS (v. 3.20.0) and using the m_map package as downloaded from https://www.eoas.ubc.ca/~rich/map.html in November 2021.

### Reporting summary

Further information on research design is available in the Nature Portfolio Reporting Summary linked to this article.

## Data availability

The Seaglider, the biogeochemical and the taxonomical data from the scientific campaign can be found at the Norwegian Polar Data Centre (Norwegian Polar Institute): https://data.npolar.no/dataset/ab96f43d-b813-4457-a466-6982c2c60a6b, https://data.npolar.no/dataset/28fbddd2-0fb2-41c9-9f42-60146e28617f and https://data.npolar.no/dataset/283e500c-732b-4f9b-a48a-3bc4990e3f55. In addition, the marine birds and mammals distribution data can be found at https://data.npolar.no/dataset/5168ad7f-4733-45fd-87bd-4c4c4c217876. Atmospheric reanalyses can be obtained from https://www.ecmwf.int/en/forecasts/datasets/reanalysis-datasets/era-interim. Satellite derived ocean color is available at https://oceandata.sci.gsfc.nasa.gov/ and https://www.eumetsat.int/ocean-colour-services. Satellite derived sea ice concentration is available at http://nsidc.org. The SOCCOM data are available at https://soccom.princeton.edu. The GLODAP Version 2 data product can be obtained from https://www.glodap.info/. Bird tracking data (2012-2016) can be found at https://www.datarepository.movebank.org/handle/10255/move.566.

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

## Acknowledgements

The research cruise to the eastern Weddell Gyre onboard the RV *Kron-prins Haakon* was funded by the Norwegian Polar Institute, with additional support from the Norwegian Ministry of Foreign Affairs. This study was also funded by the Research Council of Norway (RCN) (grant number 288370) and the National Research Foundation, South Africa (Grant UID 118715) through the Norway-South Africa collaborative project "Southern Ocean phytoplankton community characteristics, primary production, $CO_2$ flux and the effects of climate change (SOPHY-CO2)" within the SANOCEAN framework (A.F., S.J.T., S.M., T.R.-K., L.d.S., M.C., P.M., M.A.). This study was also inspired by the Research Council of Norway (RCN) funded project "I-CRYME: Impact of CRYosphere Melting on Southern Ocean Ecosystems and biogeochemical cycles" (grant number 335512, S.M., T.H., P.A., M.A., A.F., M.C.) and the Norwegian Centre of Excellence "iC3: Center for ice, Cryosphere, Carbon and Climate" (grand number 332635, PA, SM, TH). The taxonomic analyses were supported by the Polish Ministry of Science and Higher Education (W37/Svalbard/2020, J.W., M.R.). Through the use of SOCCOM data, this work was also sponsored by NSF's Southern Ocean Carbon and Climate Observations and Modeling (SOCCOM) Project under the NSF Awards PLR-1425989 and OPP-1936222, with additional support from NOAA and NASA. Logistical support for this project in the Antarctic was provided by the U.S. National Science Foundation through the U.S. Antarctic Program. The Argo float data used for the mixed layer depths calculation were collected and made freely available by the Coriolis project (http://www.coriolis.eu.org), the International Argo Project (http://www.argo.net), and the national programs that contribute to them. We thank Isabelle Giddy for helping with the analysis of mixed layer depths in the area of the bloom.

## Author contributions

All authors provided data and ideas and contributed to writing the paper. By specialty, authors contributed as follows: phytoplankton (S.M., H.K., P.A., A.Le., J.W., M.R., I.P.), physical oceanography (T.H., L.d.S., N.St., T.D.), phytoplankton physiology (T.R.-K., A.S., S.J.T.), iron (M.A., N.Sa., A.R.), chemical oceanography (A.F., M.C., P.M., O.H.), krill (T.F., E.H., J.-H.S.), birds & whales (H.S., S.D., A.Lo., H.A., E.G., N.L., A.v.T.), Cruise leader (H.S.).

## Competing interests

The authors declare no competing interests.
