## [Peer Review File · Nature Communications]

Wind-driven upwelling of iron sustains dense blooms and food webs in the eastern Weddell GyreREVIEWER COMMENTS

Reviewer #1 (Remarks to the Author):

Review of “Wind-driven upwelling of iron sustains dense phytoplankton blooms and productive food webs in the eastern Weddell Gyre” by Moreau et al.

Summary of the study: The present manuscript presents a novel description of a spring phytoplankton bloom in the Weddell Gyre observed from a combination of in situ ship, glider, and BGC-Argo float data, as well as satellite information. The authors highlight the intensity of the bloom in terms of its very high concentration of particulate organic carbon (POC) and chlorophyll (Chl), and its impact on the regional food web, inferred from cruise survey data. The bloom is attributed to bottom-up iron fertilization from hydrothermal sources based on knowledge of the region topographic features and ancillary isotopic helium data.

General assessment: Overall, I have a positive appreciation of this work. The manuscript is well written and organized. The authors present various lines of evidence to substantiate their conclusions regarding the mechanistic drivers of the bloom and its ecological implications for the regional marine food web. Given the remote nature of the Weddell Gyre and the difficulty to conduct biogeochemical research in this region due to seasonally-limited remote sensing and scarce in situ data, I think that the description of this bloom represents an important contribution to Antarctic marine research and the mechanisms that drive biogeochemical cycles in this region. The reported concentration of POC is indeed high for open ocean conditions and could represent an important source of organic chemical energy, attracting zooplankton and large marine animals and seabirds, as suggested by data collected by the authors as well as other ancillary information. Also, the upwelling of hydrothermal iron is a relatively novel mechanism of nutrient supply that deserves attention as a potentially important constraint driving food webs and regulating carbon fluxes in the Southern Ocean.

However, I do think the manuscript needs some improvement in the presentation/quality of some of its figures, and the description of the estimation of iron utilization by phytoplankton. It also seems that the materials and methods are all located on the supplementary material. It would help to move some of the more relevant methods to the end of the main manuscript. I detail these concerns and others below within the more specific comments.

Specific comments:

Abstract: In line 35 you say: “ *However, satellite chlorophyll a imagery shows that blooms of similar extent have occurred in the region in 9 out of the 22 years of available ocean color satellite data (1997-2019).*” But in line 44 “ *the extent of the bloom magnitude has never before been reported.*” This is confusing because it sounds as if the 2019 bloom is unique, which does not seem to be based on the satellite record. Perhaps simply substituting “reported” for “described” at the very end of the abstract would help.

Line 59. Is the 58 Tg C/yr estimate also for a 2000 m depth horizon? Please clarify.

Line 83 and Figure 1: “A subsurface Chl a and POC maximum was observed at the base of the ML southwards of 68.32°S (Figure 1c) where Chl a reached 1.9 mg m^{-3} and POC reached 1080 mg m^{-3} .”: It is very difficult to appreciate this feature in the vertically-resolved transects shown in Figure 1 c) and d). Consider changing the colormap and/or plotting the data in a log scale (but preserving the original concentration numbers in the colorbar tick labels). The same comment applies to figure S1.

Figure 1 a): The colorbar numbers are too small and its range is not appropriate for the dynamic range of the chlorophyll concentration shown in the figure. Also, the integrated POC markers are very difficult to read. Perhaps try using a black borderline around them. I would also consider zooming the figure nearer to the bloom area to make both the POC markers and the inverted black triangles easier to see.

Line 92: It is unclear if you only used MODIS or also Sentinel satellite data in this part of the analysis.

Line 118: Indicate what percentage do the 34 profiles with high POC represent with respect to all the available float profiles. This will help highlight the uniqueness of the bloom.

Line 127: “Therefore, the production of such striking levels of organic matter in the open waters of the Southern Ocean must have involved an important iron source.” I agree that iron seems like the most likely driver of this bloom, but you should also discuss the potential role of reduced top-down control, or how the likely rapid delivery of iron allows phytoplankton to temporarily “scape” the grazing pressure. If grazers were able to match the speed of the division rate, then it would not matter how much iron is added, one would not see the high accumulation of phytoplankton biomass appreciated in the bloom given that the steady-state balance between grazing and division rate would not be altered. See “Student's tutorial on bloom hypotheses in the context of phytoplankton annual cycles” (<https://onlinelibrary.wiley.com/doi/full/10.1111/gcb.13858>)

Line 133: “ Unfortunately, dissolved iron concentrations were not measured directly inside the open ocean bloom area during the campaign and these concentrations should only be considered representative of the broader area.”: So iron concentrations were likely higher at the core of the bloom?

Lines 135 – 143 and the overall calculation of iron demand: Did you assume that POC was all composed of phytoplankton carbon for the purpose of this calculation? A more realistic approach could be to assume that phytoplankton represents about 20 to 30 % of total POC (see Graff et al., 2015 - <https://doi.org/10.1016/j.dsr.2015.04.006>, and Arteaga et al., 2016 - <https://doi.org/10.1002/2016GB005458>). I find this part of the main manuscript and the section of the methods describing NPP and iron demands not very clear. I suggest carefully detailing and expanding on each step and assumption taken for this calculation.

Lines 146 – 153: Was this analysis carried out with 8-day composites as indicated in the methods? Also, you report two different estimates of chlorophyll, those from the glider and those from the ocean color satellite data. Did you check if they are consistent?

Figure 2: What percentage of the analyzed 8-day satellite data ended up in the composite for “open ocean bloom” (panel b) and the “No open ocean bloom” conditions (panel c).

Line 227 and Figure 3: *“However, in the predominantly westward coastal flow regime south of 65°S (yellow arrow in Figure 3), sub-surface $\delta^3\text{He}$ ratios are higher at the Prime Meridian than at 30°E (Figure 3, left panel), indicating an additional source of hydrothermally enriched waters between these two sections that are located upstream and downstream of the bloom region.”*: I am not sure that the feature described above is evident in Figure 3. Overall, it is quite difficult to digest the information in this figure, especially in the top map. I am not sure what the best solution is here, but a different way of presenting these data should be considered.

Figure 4. Similarly as for figure 1, the scale of the colorbar does not seem appropriate. The numbers of the legends in panels a) and b) are too small. In panel c), I do not think it is a good idea to plot a heatmap on top of the chlorophyll colormap, as the whole plot becomes very confusing, especially when similar colors are used in both colormaps. The same concern applies to Figure S6.

Reviewer #2 (Remarks to the Author):

**Wind-driven upwelling of iron sustains dense phytoplankton blooms and productive food webs in the eastern Weddell Gyre
Sebastien Moreau et al.**

The research links a large late summer phytoplankton bloom in the Weddell Gyre to upwelling of iron enriched deep water with a potential hydrothermal source. Bloom occurrence over the past 20 years is assessed using remote sensing. The occurrence/intensity of this bloom is linked to wind pattern changes. The presumed hydrothermal iron source are hypothesized to be located upstream of the bloom. The bloom is investigated for chlorophyll a and carbon biomass, and microscopic identification of major phytoplankton species (e.g. diatoms). Furthermore the bloom is linked to Krill distributions, and those of other top predators such as Petrels and Hump Back wales, all of which were significantly higher in the bloom area. Although the described pathways for bloom development are convincing, iron concentrations in the bloom and upstream of the bloom were not/poorly quantified. I agree with the proposed iron sources as described here, noting that this is a missing link that warrants further investigation.

The correlations between wind patterns and are significant but also not extremely strong. This leaves room for other factors that contribute to the intensity of the observed bloom. Apart from trace metal availability light is known to have significant influence in southern Ocean phytoplankton bloom development. Seasonal changes in mixed layer depth may also contribute to the intensity of the bloom in the Weddell Gyre. I wonder if the authors have investigated this. Or are the weak correlations the result of insufficient data quality?

The attractiveness of this work is that it is in general very complete, combining research on temporal scale of 20 years (remote sensing), cruise data, climate, and hydrography. In addition, it links to observations of higher trophic levels relevant for the Southern Ocean.

Least worked out (not quantified) is carbon storage in the deep ocean (line 282, Phytoplankton community composition and potential for carbon export), which remains consequently hypothetical. Although the presence of the large diatom *Chaetoceros dictyota* indeed points to a substantial carbon flux to the deep ocean.

The manuscript is well written. It provides novel insights in the development of large phytoplankton blooms in the Weddell Gyre. These blooms are important for higher trophic levels and carbon storage in the deep ocean. The manuscript can be published after minor revisions.

Detailed comments:

Figure 2: Climatic conditions inducing the open ocean bloom. Figure title does not correspond to the figure content.

Reviewer #3 (Remarks to the Author):

Moreau et al investigates the magnitude and variability of primary production in the Weddell Gyre, which has important implications for the global carbon cycle and marine ecosystems. The authors have brought together an impressive number of different datasets and clearly thought deeply about many aspects of the system. If anything, there are almost too many different threads being pursued for one manuscript. The first thread is about interannual variability of the bloom and its link to wind anomalies, the second thread is about the possible importance of hydrothermal iron as a nutrient source for the region, the third thread is about phytoplankton community composition and carbon export from the bloom, and the fourth thread is about the implications of the bloom for upper trophic level organisms.

Each of these different threads is interesting in its own right and the authors have

clearly done an immense amount of work, but the paper reads (to me) as very disjointed. The manuscript is quite long, and by the end, it's not clear what the main message of the paper even is. The title seems to imply that threads one and four (that I described above) are the focus, but this gets lost amidst all the other details being presented. In a different journal, this might be fine, but in order to be appropriate for Nature Communications, I think the manuscript needs to be condensed and streamlined to convey a stronger message. Personally, I think that the hydrothermal iron section is the part that should be shortened. But, of course. It's up to the authors to decide which of the results they think are the most important.

At any rate, most of my comments are related to the presentation of the results rather than the scientific content, although I do think some more care is needed in defining the bloom presence and mixed layer depth (MLD) variability needs to be considered more thoroughly (I explain what I mean by this later on). Overall, I think there's a lot of nice work here! More detailed comments are given below:

Lines 33-35: very minor suggested rephrasing: "Over its two and a half months duration, this phytoplankton bloom accumulated up to 20 g C m⁻² of organic matter, which is unusually high for Southern Ocean open waters."

Line 35: I don't understand why "However" is an appropriate connector here, I would simply delete it and start the sentence "Satellite chlorophyll..."

Line 39-41: I think this is the first hint that there are too many different threads in the paper, because this sentence could be deleted without affecting the flow of the abstract at all. Why does it matter (to the rest of the results) that the iron is hydrothermally-sourced? It's not clear based on how the abstract is written currently.

Line 42-44: I don't think this is a very strong concluding sentence for the abstract. The final sentence (in my view) should say something more explicit about the implications of the results, rather than just saying that the bloom magnitude has never been reported before. So what? Why does it matter that the bloom magnitude has never been reported?

Line 50: this is a small quibble, but I don't really agree that the Weddell Gyre is a "representative region" of the Southern Ocean. In fact, I'd argue that it's pretty different dynamically from the rest of the Antarctic margins (except for the Ross Gyre).

Lines 82-83: the fact that the bloom was subsurface intensified is never really followed up on. Is this perhaps because the measurements were collected towards the end of the bloom after surface nutrients had already been depleted?

Lines 116-117: I think this sentence can be deleted and then the following sentence could begin "We identified only..."

Lines 129-132: It seems surprising that iron is not limiting so late in the season, especially since the chlorophyll and POC were strongly subsurface intensified. Why would productivity occur so close to the base of the mixed layer if light were limiting?

Lines 148-150: This definition of the bloom doesn't account for the fact that the bloom position could potentially shift from year to year. Based on this calculation alone, you cannot rule out the possibility that the bloom magnitude is roughly the same each year but that it's position changes (moving the core of the bloom out of your box in certain years). More justification needs to be given for these box boundaries.

Lines 150-153: What about the length of the bloom? How much does that change from year to year?

Lines 170-172: Some mention of the MLD seems absent here. The picture from Tagliabue

et al. (2014) is that most of the surface iron resupply occurs through mixed-layer entrainment in winter. It would be possible then, in theory, that the bloom magnitude is linked to the depth of the maximum MLD in the preceding winter. This could also be related to the anomalies in wind-driven upwelling, since the enhanced upwelling would presumably raise the depth of the ferricline, which would then increase the entrainment flux of iron. It might be worth quickly trying to assess interannual variability in MLD from Argo data or from some model. Or at any rate, at least mention MLD somewhere in this discussion.

Lines 187-190: I don't understand the logic here because the iron is typically not limiting at the time of bloom initiation (since it has been resupplied through wintertime mixing). Usually we might expect a progression from light limitation early in the season to nutrient limitation later in the season. If this is the case, upwelling shouldn't necessarily trigger the bloom (although it's obviously important in sustaining it). von Berg et al. (2020) showed a strong relationship between bloom initiation and the timing of sea ice retreat in this region.

Lines 197-198: Yes, but in that study, "upwell" referred to crossing 1000 m, which is not necessarily synonymous with where waters reach the surface ocean (which is what we actually care about for phytoplankton growth).

Lines 218-222: I think this statement is too strong, the isotopic Helium measurements support hydrothermalism as an iron source, but I don't see how these data show that it is the "main" iron source.

Lines 255-256: I don't see how you can rule out continental shelf-sourced iron. Although the bloom is far from the coast, iron could be advected offshore below the mixed-layer and then brought up to the surface in the bloom region by the same wind-driven upwelling discussed previously.

Lines 258-259: satellite ocean color data cannot be retrieved when sea ice is present, so naturally the satellite data will show that the bloom occurred when sea ice was absent. But this does not rule out the possibility of under-ice blooms.

Lines 264-266: Similar to my comment above, sedimentary iron could in theory be advected long distances beneath the mixed-layer and then brought up to the surface in the bloom region by the upwelling in the gyre.

Lines 271-273: I don't understand the reasoning for ruling out entrainment. As stated in the following line, even if the iron supplied by entrainment is utilized, subsequent iron recycling could potentially help sustain the bloom.

Lines 276-279: As I said at the start, I feel there is too much focus put on the potential hydrothermal origin of the iron. At this point in reading the paper, the main message has been muddled and I think this whole section could be condensed. Or else, it needs to be better clarified what the significance is (to the rest of the paper) that the iron is of hydrothermal origin.

Lines 281-311: Similar to my point above, the main message of the paper has been lost somewhat at this point in the manuscript. There is a lot of information in this section, but it's not clear to me what purpose it serves to the broader story.

Lines 314-325: Sorry to keep repeating myself, but again, there's tons of information in this paragraph, but I think you need to better establish at the start of the section how this information fits into the broader story. This feels disjointed from the previous section, which in turn, felt disjointed from the section before that.

Lines 342-392: Macrofauna are not my area of specialty, so I can't comment much on the scientific content in section, but I can say that it reads like a completely different

paper to me. As above, I think the main message of the paper needs to be better clarified and then this information needs to be better integrated into that message.

Lines 395-408: I think the conclusion section will be strengthened once the main message of the paper is better established. Note that although a significant portion of the paper is dedicated to discussing the hydrothermal origin of the iron, it is not mentioned anywhere in this section, which seemingly confirms my impression that it is not as crucial to the story being told here to warrant so much space in the manuscript.

Figure 1: It seems strange to me to use a cyclical, nonlinear colormap for chlorophyll and the integrated POC points are not visible against the underlying basemap.

Change the color of the MLD line (maybe black?) as it's hard to see against the POC colormap and panel c) is not visible to people with red-green colorblindness.

Figure 4: Again, this colormap is odd to me for chlorophyll. No legend for the heat map in panel c)

Response to the reviewers

We answer below each point raised by the three reviewers. Our answers are written in blue. Text from the manuscript is copied in italic and the modified parts of the manuscript are in italic and underlined.

REVIEWER COMMENTS

Reviewer #1 (Remarks to the Author):

Review of “Wind-driven upwelling of iron sustains dense phytoplankton blooms and productive food webs in the eastern Weddell Gyre” by Moreau et al.

Summary of the study: The present manuscript presents a novel description of a spring phytoplankton bloom in the Weddell Gyre observed from a combination of in situ ship, glider, and BGC-Argo float data, as well as satellite information. The authors highlight the intensity of the bloom in terms of its very high concentration of particulate organic carbon (POC) and chlorophyll (Chl), and its impact on the regional food web, inferred from cruise survey data. The bloom is attributed to bottom-up iron fertilization from hydrothermal sources based on knowledge of the region topographic features and ancillary isotopic helium data.

General assessment: Overall, I have a positive appreciation of this work. The manuscript is well written and organized. The authors present various lines of evidence to substantiate their conclusions regarding the mechanistic drivers of the bloom and its ecological implications for the regional marine food web. Given the remote nature of the Weddell Gyre and the difficulty to conduct biogeochemical research in this region due to seasonally-limited remote sensing and scarce in situ data, I think that the description of this bloom represents an important contribution to Antarctic marine research and the mechanisms that drive biogeochemical cycles in this region. The reported concentration of POC is indeed high for open ocean conditions and could represent an important source of organic chemical energy, attracting zooplankton and large marine animals and seabirds, as suggested by data collected by the authors as well as other ancillary information. Also, the upwelling of hydrothermal iron is a relatively novel mechanism of nutrient supply that deserves attention as a potentially important constraint driving food webs and regulating carbon fluxes in the Southern Ocean.

However, I do think the manuscript needs some improvement in the presentation/quality of some of its figures, and the description of the estimation of iron utilization by phytoplankton. It also seems that the materials and methods are all located on the supplementary material. It would help to move some of the more relevant methods to the end of the main manuscript. I detail these concerns and others below within the more specific comments.

We thank the reviewer for the very positive comments on our manuscript. We answer below each of the points raised by the reviewer.

In addition, the reviewer is right and the section previously entitled as “Supplementary Materials and Methods” corresponds to the “Methods” section that should come at the end of the main manuscript, following the style of articles published in Nature Communications. We modified this in the new version of the manuscript.

Specific comments:

Abstract: In line 35 you say: “*However, satellite chlorophyll a imagery shows that blooms of similar extent have occurred in the region in 9 out of the 22 years of available ocean color satellite data (1997-2019).*” But in line 44 “*the extent of the bloom magnitude has never before been reported.*” This is confusing because it sounds as if the 2019 bloom is unique, which does not seem to be based on the satellite record. Perhaps simply substituting “reported” for “described” at the very end of the abstract would help.

This is a very good suggestion. As Reviewer #3 also had a comment on the last sentence of the abstract, we decided to remove it, however.

Line 59. Is the 58 Tg C/yr estimate also for a 2000 m depth horizon? Please clarify.

This estimate of the Weddell Gyre carbon dioxide (CO₂) sink of ~50 Tg C/yr corresponds to the carbon enrichment of the CDW that leaves the Weddell Gyre underneath the ACC as described in MacGillchrist et al. (2019). We provide clarity in this section as follows and we refer to the study of MacGillchrist et al. (2019) for further details:

“A recent carbon budget across the Weddell Gyre¹, based on observations of inorganic carbon, suggests that intense primary productivity in the region of the Weddell Gyre, away from continental shelves, can explain the carbon drawdown estimated from the carbon deficit directed out of the Gyre². The strength of the resulting Weddell Gyre carbon dioxide (CO₂) sink to depth, at an estimated ~50 Tg C yr⁻¹, is significant when compared to the global ocean biological carbon pump, estimated at 430 Tg C yr⁻¹ across the 2,000 m horizon by Honjo et al. (2008; ref. ³).”

Line 83 and Figure 1: *“A subsurface Chl a and POC maximum was observed at the base of the ML southwards of 68.32°S (Figure 1c) where Chl a reached 1.9 mg m⁻³ and POC reached 1080 mg m⁻³.”*: It is very difficult to appreciate this feature in the vertically-resolved transects shown in Figure 1 c) and d). Consider changing the colormap and/or plotting the data in a log scale (but preserving the original concentration numbers in the colorbar tick labels). The same comment applies to figure S1.

We thank the reviewer for this comment. We modified the color scale in Figure 1c and 1d and Figure S1 to use a logarithmic color scale but keeping the original concentrations numbers.

Figure 1 a): The colorbar numbers are too small and its range is not appropriate for the dynamic range of the chlorophyll concentration shown in the figure. Also, the integrated POC markers are very difficult to read. Perhaps try using a black borderline around them. I would also consider zooming the figure nearer to the bloom area to make both the POC markers and the inverted black triangles easier to see.

We considered the reviewer’s comment and modified the colorbar, increasing the font size and changing the range of concentrations shown. The reviewer was right, and the range was too large. We also added an insert with a zoom in the core area of the bloom to better show the integrated POC markers. We refer the reviewer to the new version of Figure 1a.

Line 92: It is unclear if you only used MODIS or also Sentinel satellite data in this part of the analysis.

The reviewer is correct and only MODIS data was used for this analysis. We have clarified the sentence to read as follows:

“In addition, satellite imagery showed that the bloom started in early-January (January 9th), reached its maximum concentration by mid-February (Supplementary Figure 2a), before it collapsed mid-March (March 14th) after our visit of the area⁴ (see Methods for a description of the MODIS satellite-derived ocean color data and the bloom detection criteria used for this analysis).”

Line 118: Indicate what percentage do the 34 profiles with high POC represent with respect to all the available float profiles. This will help highlight the uniqueness of the bloom.

This is a good suggestion, thank you. We modified the sentence to:

“We identified only 34 profiles (0.36% of all the SOCCOM BGC-Argo profiles studied) where the integrated POC over the upper 100 m amounted to more than 15 g C m⁻² and up to 19.7 g C m⁻² over the whole open Southern Ocean (see Methods and Supplementary Figure 3a).”

Line 127: *“Therefore, the production of such striking levels of organic matter in the open waters of the Southern Ocean must have involved an important iron source.”* I agree that iron seems like the most likely driver of this bloom, but you should also discuss the potential role of reduced top- down control, or how the likely rapid delivery of iron allows phytoplankton to temporarily “scape” the grazing pressure. If grazers were able to match the speed of the division rate, then it would not matter how much iron is added, one would not see the high accumulation of phytoplankton biomass appreciated in the bloom given that the steady-state balance between

grazing and division rate would not be altered. See “Student's tutorial on bloom hypotheses in the context of phytoplankton annual cycles” (<https://onlinelibrary.wiley.com/doi/full/10.1111/gcb.13858>)

This is a very good point raised by the reviewer. We have modified the following sentence to acknowledge for the potential impact of grazing in the Southern Ocean:

“Therefore, the production of such striking levels of organic matter in the open waters of the Southern Ocean must have involved an important iron source and a relatively low grazing pressure at the onset of the bloom as grazing can dominate phytoplankton loss in the Southern Ocean”.

Line 133: “Unfortunately, dissolved iron concentrations were not measured directly inside the open ocean bloom area during the campaign and these concentrations should only be considered representative of the broader area.”: So iron concentrations were likely higher at the core of the bloom?

This is a possibility indeed, one that warrants future research in the area where these blooms develop. Unfortunately, we do not have the data to add a comment on this possibility to the manuscript.

Lines 135 – 143 and the overall calculation of iron demand: Did you assume that POC was all composed of phytoplankton carbon for the purpose of this calculation? A more realistic approach could be to assume that phytoplankton represents about 20 to 30 % of total POC (see Graff et al., 2015 - <https://doi.org/10.1016/j.dsr.2015.04.006>, and Artega et al., 2016 - <https://doi.org/10.1002/2016GB005458>). I find this part of the main manuscript and the section of the methods describing NPP and iron demands not very clear. I suggest carefully detailing and expanding on each step and assumption taken for this calculation.

This is a very good point raised by the reviewer. Not all the POC measured inside the bloom was of autotrophic origin. While the reviewer is right that phytoplankton probably represents 20 to 30% of the total POC, we pointed to this fact in the previous version of the manuscript with the following sentence:

“Uncertainties on the lower and upper limit of this range are large as they do not account for the role of grazing or vertical export on POC in the ML⁵, and as POC is both constituted of autotrophic and heterotrophic carbon.”

To acknowledge the reviewer’s comment, we added the following two sentences, and a reference to the Artega et al. (2016) study, to the section of the manuscript where we discuss iron demand:

“We calculated the iron necessary to accumulate POC levels up to 20 g C m⁻² to be between 3.8 and 33.3 μmol Fe m⁻² by using lower and upper bound Fe:C ratios from Fe-replete and Fe-deplete Southern Ocean waters (2.3 to 20 mol:mol 10⁻⁶)⁶. Uncertainties on the lower and upper limit of this range are large as they do not account for the role of grazing or vertical export on POC in the ML⁵, and as POC is both constituted of autotrophic and heterotrophic carbon. A lower estimate could be obtained by considering that phytoplankton only represents 20 to 30% of the total POC⁷. With this consideration, the necessary iron to accumulate 5 g C m⁻² (i.e., 25% of 20 g C m⁻²) was calculated to be between 1 and 8.3 μmol Fe m⁻². An estimation of satellite-derived net primary productivity and the associated iron demands during the 2.5 months bloom period ranged between 14.8 and 42.6 μmol Fe m⁻² (see Methods). Together, these observations and calculations highlight that an important iron source or mechanism was required in order to lead to such high levels of POC accumulation inside the open ocean bloom.”

Lines 146 – 153: Was this analysis carried out with 8-day composites as indicated in the methods? Also, you report two different estimates of chlorophyll, those from the glider and those from the ocean color satellite data. Did you check if they are consistent?

Yes, this analysis was carried out with the 8-day composites, i.e. 8-day means, as indicated in the Methods. We now clarify this with the following sentence in the Methods:

“First, the long-term analysis of the bloom presence (Figure 2a) was performed from January 1997 to December 2020, covering the ocean-color satellite era. We defined the bloom presence when the 8-day mean satellite-

derived Chl a, averaged over 4°-8°E and 67.8°-68.4°S, was larger than the 23-year long mean Chl a + 1 standard deviation over that area (i.e., 1.14 mg m⁻³, Figure 2a)."

To answer the second part of the reviewer's comment, we compared the Chl a concentration at 10 m depth from the glider profiles with Satellite derived Chl a concentration (MODIS level 3 Chl a data processed with the default chlorophyll algorithm (*chlor_a*) at the corresponding date, latitude and longitude. The Chl a concentration at 10 m depth averaged 0.57 ± 0.04 mg m⁻³ for all the glider profiles. The corresponding Chl a concentration from MODIS was in the same range and averaged 0.46 ± 0.01 mg m⁻³. Thus, the two datasets can thus be considered reasonably consistent.

Figure 2: What percentage of the analyzed 8-day satellite data ended up in the composite for "open ocean bloom" (panel b) and the "No open ocean bloom" conditions (panel c).

As described in the Methods, we computed de-seasoned anomalies of monthly Chl a, east-west winds and sea ice concentration by first removing the monthly climatology from monthly means, producing de-seasoned anomalies from September 1997 to December 2019. Thereafter, we produced composite maps by averaging monthly de-seasoned anomalies during and outside of the bloom period (i.e. with the bloom period as defined above). The composite map during the bloom presence represents 16% of the total monthly de-seasoned anomalies. We added the following sentence to the new version of the manuscript to address this point:

"The composite map during the bloom presence represents 16% of the total available (monthly de-seasoned) values."

Line 227 and Figure 3: *"However, in the predominantly westward coastal flow regime south of 65°S (yellow arrow in Figure 3), sub-surface $\delta^3\text{He}$ ratios are higher at the Prime Meridian than at 30°E (Figure 3, left panel), indicating an additional source of hydrothermally enriched waters between these two sections that are located upstream and downstream of the bloom region."*: I am not sure that the feature described above is evident in Figure 3. Overall, it is quite difficult to digest the information in this figure, especially in the top map. I am not sure what the best solution is here, but a different way of presenting these data should be considered.

As the reviewer suggested, we have considered several ways to present this figure. To best represent the message given in the paper, we added an insert zooming in the area of the $\delta^3\text{He}$ profiles transects, thereby clarifying the positions of the $\delta^3\text{He}$ profiles shown in the three transects. We think this change addresses the reviewer's concern and we refer the reviewer to the new version of the Figure 3.

Figure 4. Similarly as for figure 1, the scale of the colorbar does not seem appropriate. The numbers of the legends in panels a) and b) are too small. In panel c), I do not think it is a good idea to plot a heatmap on top of the chlorophyll colormap, as the whole plot becomes very confusing, especially when similar colors are used in both colormaps. The same concern applies to Figure S6.

Similarly to Figure 1a, we modified the colorbar used in Figures 4 and S2, increasing the font size and changing the range of concentrations shown. In addition, we removed the background Chlorophyll colormap from the panel c) of Figure 4, as suggested by the reviewer. We also did it for all panels in Figure S6.

We refer the reviewer to the new version of Figure 4, S2 and S6.

Reviewer #2 (Remarks to the Author):

Wind-driven upwelling of iron sustains dense phytoplankton blooms and productive food webs in the eastern Weddell Gyre

Sebastien Moreau et al.

The research links a large late summer phytoplankton bloom in the Weddell Gyre to upwelling of iron enriched deep water with a potential hydrothermal source. Bloom occurrence over the past 20 years is assessed using

remote sensing. The occurrence/intensity of this bloom is linked to wind pattern changes. The presumed hydrothermal iron source are hypothesized to be located upstream of the bloom.

The bloom is investigated for chlorophyll a and carbon biomass, and microscopic identification of major phytoplankton species (e.g. diatoms). Furthermore the bloom is linked to Krill distributions, and those of other top predators such as Petrels and Hump Back wales, all of which were significantly higher in the bloom area.

Although the described pathways for bloom development are convincing, iron concentrations in the bloom and upstream of the bloom were not/poorly quantified. I agree with the proposed iron sources as described here, noting that this is a missing link that warrants further investigation.

The correlations between wind patterns and are significant but also not extremely strong. This leaves room for other factors that contribute to the intensity of the observed bloom. Apart from trace metal availability light is known to have significant influence in Southern Ocean phytoplankton bloom development. Seasonal changes in mixed layer depth may also contribute to the intensity of the bloom in the Weddell Gyre. I wonder if the authors have investigated this. Or are the weak correlations the result of insufficient data quality?

The attractiveness of this work is that it is in general very complete, combining research on temporal scale of 20 years (remote sensing), cruise data, climate, and hydrography. In addition, it links to observations of higher trophic levels relevant for the Southern Ocean.

Least worked out (not quantified) is carbon storage in the deep ocean (line 282, Phytoplankton community composition and potential for carbon export), which remains consequently hypothetical. Although the presence of the large diatom *Chaetoceros dichaeta* indeed points to a substantial carbon flux to the deep ocean.

The manuscript is well written. It provides novel insights in the development of large phytoplankton blooms in the Weddell Gyre. These blooms are important for higher trophic levels and carbon storage in the deep ocean. The manuscript can be published after minor revisions.

We thank the reviewer for the very positive and constructive comments on our manuscript. The reviewer is right to point to light and changes in the mixed layer depth as potential important contributors to phytoplankton bloom development in the Southern Ocean. We further developed these aspects in the new version of the manuscript.

First, regarding light, we modified the following sentence to state that light was not limiting primary productivity in the study area at the time of our campaign:

"During our campaign, light and macronutrients were not found to be limiting throughout the open ocean bloom (as shown in Kauko et al. 2021, ref. 4)."

In addition, we would like to refer the reviewer to this paragraph from the original version of the manuscript where we discuss relief from light limitation following ice retreat in the spring.

"Furthermore, at times when the open ocean bloom takes place, the stronger easterlies push sea ice south, towards the coast. This leads to a negative anomaly in sea-ice concentration in the open ocean west of Astrid Ridge and the opposite positive anomaly in sea-ice concentration along the coast between Astrid Ridge and the Fimbul Ice Shelf Tongue (Figure 2f). By pushing the sea ice south, towards the coast, enhanced easterlies may also relieve phytoplankton production from possible light limitation by the sea ice cover³."

Second, the reviewer is right and the maximum winter MLD could be playing an important role in setting the stage for the following summer phytoplankton bloom, as described in Tagliabue et al. (2014) and as was also suggested by Reviewer #3.

To test this hypothesis, we gathered Argo floats data from this area from 2007 to 2019 and determined the mixed layer depths from all profiles, thus building a long-term time series of mixed layer depths in the area. We

built this time series from 2007 to 2019 since, before 2007, the Argo floats seasonal coverage was insufficient in this area. We added the method description of this new analysis to the new version of the Methods section:

“Determination of mixed layer depth in the area

We calculated the mixed layer depth in the area of the bloom between 2002 and 2020 by using all Argo floats that drifted through the area within 4 to 8E and -69 to -67S, i.e., total of 1024 profiles of temperature, salinity and depth. Furthermore, we removed profiles earlier than 2007 due to large seasonal gaps. From all profiles, we calculated the mixed layer depth as the depth where the potential density exceeds its 10 m value by 0.03 km/m³ (de Boyer Montegut et al. 2004, ref. ⁹). We then averaged the obtained mixed layer depths to monthly means.”

We then used this time series to test the hypothesis that the bloom magnitude was linked to the depth of the maximum MLD in the preceding winter. We separated the years during which a high biomass bloom took place from all other years. For this, we considered years starting from July 1 and ending on June 30 to include the austral summer primary production. We found that there was no significant difference in maximum winter MLD between years with and years without a high biomass bloom. To consider the reviewer’s comment and this new analysis, we added the following text to the manuscript as well as a new panel to Supplementary Figure 4e:

“Entrainment by convective mixing during winter has been highlighted as another mechanism that can make iron from deeper water masses available to the surface with fluxes of up to 33.2 $\mu\text{mol Fe m}^{-2} \text{yr}^{-1}$ being observed¹⁰. Based on existing Argo float profiles, we tested the hypothesis that the bloom magnitude was linked to the preceding winter maximum ML depth from 2007 to 2019 (see Methods). We found that no significant differences existed in preceding winter (i.e. September) maximum ML depths between years with (mean of 124.9 \pm 6.3 m) and years without (mean of 113.1 \pm 13.3 m) an open ocean bloom ($p > 0.05$, Supplementary Figure 4e).”

Detailed comments:

Figure 2: Climatic conditions inducing the open ocean bloom. Figure title does not correspond to the figure content.

We modified the figure title to: *“Figure 2: Composite anomalies in the presence and absence of the open ocean bloom”*

Reviewer #3 (Remarks to the Author):

Moreau et al investigates the magnitude and variability of primary production in the Weddell Gyre, which has important implications for the global carbon cycle and marine ecosystems. The authors have brought together an impressive number of different datasets and clearly thought deeply about many aspects of the system. If anything, there are almost too many different threads being pursued for one manuscript. The first thread is about interannual variability of the bloom and its link to wind anomalies, the second thread is about the possible importance of hydrothermal iron as a nutrient source for the region, the third thread is about phytoplankton community composition and carbon export from the bloom, and the fourth thread is about the implications of the bloom for upper trophic level organisms.

Each of these different threads is interesting in its own right and the authors have clearly done an immense amount of work, but the paper reads (to me) as very disjointed. The manuscript is quite long, and by the end, it’s not clear what the main message of the paper even is. The title seems to imply that threads one and four (that I described above) are the focus, but this gets lost amidst all the other details being presented. In a different journal, this might be fine, but in order to be appropriate for Nature Communications, I think the manuscript needs to be condensed and streamlined to convey a stronger message. Personally, I think that the hydrothermal iron section is the part that should be shortened. But, of course. It’s up to the authors to decide which of the results they think are the most important.

At any rate, most of my comments are related to the presentation of the results rather than the scientific content, although I do think some more care is needed in defining the bloom presence and mixed layer depth (MLD) variability needs to be considered more thoroughly (I explain what I mean by this later on). Overall, I think there’s a lot of nice work here! More detailed comments are given below:

We thank the reviewer for the positive comments about our paper and we are also thankful for the constructive criticism given. We give below detailed answers to all the comments raised by the reviewer.

Regarding the overall length of the paper and the organisation of the various threads, we would like to keep the manuscript as much as possible as we originally designed it. We believe that the combination of the various datasets that we have used to build our argument around this paper are rare, with consistent evidence being drawn from physical oceanography and iron sources, which drive an outstanding phytoplankton accumulation in an area that also attracts a rich Southern Ocean ecosystem. Such an ecosystem feature is only encountered rarely when going on an oceanographic cruise. Each line of evidence seems to play an important part in the greater picture. With this paper, we want to provide a comprehensive description of this phenomenon, its driving mechanisms and its biogeochemical and ecological consequences. For these reasons, we would like to keep the paper's current organisation. However, we recognize the concerns of the reviewer that the paper felt "disjointed" and in response we have worked on reducing some sections and on improving the connections between the various threads throughout the paper to refine the flow of the manuscript.

Moreover, we have followed the reviewer's recommendations and have further considered other iron sources that may have also contributed to the sustained large open ocean bloom reported here. In addition, we have added a new analysis to test the hypothesis that the bloom magnitude was linked to the depth of the maximum mixed layer depth in the preceding winter. We refer the reviewer to our detailed responses below. We believe that these changes improved substantially the flow of the paper and we hope the reviewer will appreciate positively this new version of the manuscript.

1. Lines 33-35: very minor suggested rephrasing: "Over its two and a half months duration, this phytoplankton bloom accumulated up to 20 g C m⁻² of organic matter, which is unusually high for Southern Ocean open waters."

We have modified the sentence as suggested.

2. Line 35: I don't understand why "However" is an appropriate connector here, I would simply delete it and start the sentence "Satellite chlorophyll..."

Following our modifications of the abstract, we have removed this sentence from the new version of the manuscript.

3. Line 39-41: I think this is the first hint that there are too many different threads in the paper, because this sentence could be deleted without affecting the flow of the abstract at all. Why does it matter (to the rest of the results) that the iron is hydrothermally-sourced? It's not clear based on how the abstract is written currently.

The reviewer is right and this was not clear in the previous version of the Abstract. Following this comment and other comments made by the reviewer below (comments #11, 14-15, 17- 19), we have modified the following sentence of the abstract to clarify our statement:

"We show that, over 1997-2019, this open ocean bloom was fueled by anomalies in easterly winds that push sea ice southwards and favor the upwelling of Warm Deep Water enriched in hydrothermal iron and, possibly, other iron sources."

4. Line 42-44: I don't think this is a very strong concluding sentence for the abstract. The final sentence (in my view) should say something more explicit about the implications of the results, rather than just saying that the bloom magnitude has never been reported before. So what? Why does it matter that the bloom magnitude has never been reported?

We modified the final sentence of the abstract to better reflect the implications of the results we present in this study. The final sentence of the new version of the abstract reads:

“This recurring open ocean bloom likely facilitates enhanced carbon export and sustains high standing stocks of Antarctic krill, supporting feeding hot spots for marine birds and baleen whales.”

5. Line 50: this is a small quibble, but I don't really agree that the Weddell Gyre is a “representative region” of the Southern Ocean. In fact, I'd argue that it's pretty different dynamically from the rest of the Antarctic margins (except for the Ross Gyre).

We understand the reviewer's point. Thus, we modified the sentence to remove “representative”.

6. Lines 82-83: the fact that the bloom was subsurface intensified is never really followed up on. Is this perhaps because the measurements were collected towards the end of the bloom after surface nutrients had already been depleted?

Following this comment and the following comment #8 from the reviewer, we looked back at the data. While there is indeed a marked subsurface maximum in POC and Chl *a*, close to the base of the mixed layer, both POC and Chl *a* also showed elevated concentrations throughout the mixed layer as can be seen in Figure 1c and d.

In the previous version of the manuscript, we stated that the macronutrients were not limiting at the time of sampling and dissolved iron was elevated, and probably not limiting, at the sea surface throughout the study area. We refer the reviewer to the following text from the present and original versions of the manuscript:

“In the Southern Ocean, primary production is usually thought to be primarily limited by iron (Fe)¹¹ while the concentrations of macronutrients (nitrate, phosphate and silicic acid) are typically high and non-limiting. During our campaign, light and macronutrients were not found to be limiting throughout the open ocean bloom (as shown in Kauko et al. 2021, ref. ⁴). Therefore, the production of such striking levels of organic matter in the open waters of the Southern Ocean must have involved an important iron source and a relatively low grazing pressure at the onset of the bloom as grazing can dominate phytoplankton loss in the Southern Ocean⁵. During our campaign, dissolved iron (dFe) was relatively elevated¹², and probably not limiting, at the sea surface throughout the study area, ranging from 0.65 ± 0.2 nM at Astrid Ridge, 0.4 ± 0.1 nM at Maud Rise and 0.6 ± 0.04 nM at 2 stations studied south of the bloom we report here (Supplementary Figure 2b). Unfortunately, dissolved iron concentrations were not measured directly inside the open ocean bloom area during the campaign and these concentrations should only be considered representative of the broader area.”

Since, this POC and Chl *a* maximum is marked but that both the POC and Chl *a* show elevated concentration in the rest of the mixed layer, we believe that this may point towards the beginning of nutrient limitation and bloom demise phase, as the reviewer suggests. Since macronutrients were not limiting in the mixed layer at the time of sampling, this may point towards the beginning of iron limitation. Therefore, to consider the reviewer's comment, we modified the following sentence to:

*“A subsurface Chl *a* and POC maximum was however observed at the base of the ML southwards of 68.32°S (Figure 1c) where Chl *a* reached 1.9 mg m^{-3} and POC reached 1080 mg m^{-3} , which may indicate the beginning of nutrient limitation in the surface mixed layer.”*

And we added the following sentence to the part of the manuscript where we discuss iron concentrations and possible limitation:

*“During our campaign, dissolved iron (dFe) was relatively elevated¹², and probably not limiting, at the sea surface throughout the study area, ranging from 0.65 ± 0.2 nM at Astrid Ridge, 0.4 ± 0.1 nM at Maud Rise and 0.6 ± 0.04 nM at 2 stations studied south of the bloom we report here (Supplementary Figure 2b). Unfortunately, dissolved iron concentrations were not measured directly inside the open ocean bloom area during the campaign and these concentrations should only be considered representative of the broader area. The accumulation of both Chl *a* and POC at the base of the mixed layer and the very low quantum yield of photosystem II, F_v/F_m may, however, indicate that iron was becoming limiting at the time of sampling.”*

We believe our answer to this comment also provides an answer to the reviewer's comment #8 below.

7. Lines 116-117: I think this sentence can be deleted and then the following sentence could begin “We identified only...”

We followed the reviewer’s suggestion and deleted this sentence.

8. Lines 129-132: It seems surprising that iron is not limiting so late in the season, especially since the chlorophyll and POC were strongly subsurface intensified. Why would productivity occur so close to the base of the mixed layer if light were limiting?

We believe we responded to this comment in our response to comment #6.

9. Lines 148-150: This definition of the bloom doesn’t account for the fact that the bloom position could potentially shift from year to year. Based on this calculation alone, you cannot rule out the possibility that the bloom magnitude is roughly the same each year but that it’s position changes (moving the core of the bloom out of your box in certain years). More justification needs to be given for these box boundaries.

We added the following justification for how we define these box boundaries:

“Since the surface area of the bloom as observed from satellite-derived Chl a in March 2019 follows the deep bathymetric contours at 3500 and 4000 m depth (Figure 1a), we defined the boundaries of this part of the study as the core of the bloom as we observed it in March 2019 and in other years, averaged over 4°-8°E and 67.8°-68.4°S.”

10. Lines 150-153: What about the length of the bloom? How much does that change from year to year? We computed the bloom phenology in the bloom area (averaged over 4°-8°E and 67.8°-68.4°S) following the methodology of Kauko et al. (2021), who followed the methodology of Racault et al. (2012). Here is a description of the method as presented in Kauko et al. (2021):

“We used a relative threshold -based approach as the bloom detection method, where the threshold was set to 1.05 times the annual median, which is a commonly used method (e.g., Racault et al., 2012; and for the Southern Ocean see Thomalla et al., 2011; Soppa et al., 2016).

To determine bloom initiation (Bi), the Chl a concentration had to exceed the threshold for at least 2 consecutive weeks. Bloom end (Be) was determined when the concentration for the last time exceeded the threshold on at least 2 consecutive weeks. The “first” bloom initiation and the “last” bloom end were recorded – i.e., if the concentration would fall below the threshold in between, this was not considered (a single bloom is characteristic for the Southern Ocean annual phytoplankton cycle; Arrigo et al., 2008; Ardyna et al., 2017). The bloom duration (Bd) is the time between bloom initiation and bloom end. In addition, maximum concentration (bloom amplitude, Ba), its timing (Bt), and the average concentration during the bloom (Bm) were calculated. “

We calculated the bloom start, end and duration between the years when the open ocean bloom took place and the other years. The results of these calculations are given in Table 1 below. The bloom started and ended a few days later (5 and 7 days, respectively) and lasted a few days longer (2 days) in the years when we identified an open ocean bloom to occur. The bloom phenology was not significantly different between years with and years without an open ocean bloom ($p > 0.05$).

Table 1: Bloom phenology indices (bloom start, end, and duration) in the area of the open ocean bloom (4°-8°E and 67.8°-68.4°S) and for the years with an open ocean bloom, the years without an open ocean bloom and all years. Dates are given in days (January 1st being Day #1).

	Bloom start	Bloom end	Bloom duration
Years with open ocean bloom	18.6 ± 5	64.7 ± 4.1	46.1 ± 3.8
Years without open ocean bloom	13.7 ± 4.9	57.4 ± 4.1	43.7 ± 5.9

All years	15.2 ± 3.7	59.7 ± 3.1	44.5 ± 4.1
-----------	------------	------------	------------

To account for this comment, we added the following sentence:

“The bloom phenology in the area (calculated following Kauko et al., 2021, ref.¹⁵) was not significantly different ($p > 0.05$) between years with and years without an open ocean bloom, and typically initiated on January 15th ± 4 days, ended on March 1st ± 3 days, and lasted 44.5 ± 4 days on average.”

11. Lines 170-172: Some mention of the MLD seems absent here. The picture from Tagliabue et al. (2014) is that most of the surface iron resupply occurs through mixed-layer entrainment in winter. It would be possible then, in theory, that the bloom magnitude is linked to the depth of the maximum MLD in the preceding winter. This could also be related to the anomalies in wind-driven upwelling, since the enhanced upwelling would presumably raise the depth of the ferricline, which would then increase the entrainment flux of iron. It might be worth quickly trying to assess interannual variability in MLD from Argo data or from some model. Or at any rate, at least mention MLD somewhere in this discussion.

the reviewer is right and the maximum winter MLD could be playing an important role in setting the stage for the following summer phytoplankton bloom, as described in Tagliabue et al. (2014) and as also suggested by Reviewer #2.

To test this hypothesis, we gathered Argo floats data from this area from 2007 to 2019 and determined the mixed layer depths from all profiles, thus building a long-term time series of mixed layer depths in the area. We build this time series from 2007 to 2019 since, before 2007, the Argo floats coverage was insufficient in this area to allow us build such a time series. We added the method description of this new analysis to the new version of the Methods section:

“Determination of mixed layer depth in the area

We calculated the mixed layer depth in the area of the bloom between 2002 and 2020 by using all Argo floats that drifted through the area within 4 to 8E and -69 to -67S, i.e., total of 1024 profiles of temperature, salinity and depth. Furthermore, we removed profiles earlier than 2007 due to large seasonal gaps. From all profiles, we calculated the mixed layer depth as the depth where the potential density exceeds its 10 m value by 0.03 km/m³ (de Boyer Montegut et al. 2004, ref. ⁹). We then averaged the obtained mixed layer depths to monthly means.”

We then used this time series to test the hypothesis that the bloom magnitude was linked to the depth of the maximum MLD in the preceding winter. We separated the years during which a high biomass bloom took place from all other years. For this, we considered years starting from July 1 and ending on June 30 to include the austral summer primary production. We found that there was no significant difference in maximum winter MLD between years with and years without a high biomass bloom. To consider the reviewer’s comment and this new analysis, we added the following text to the manuscript as well as a new panel to Supplementary Figure 4e:

“Entrainment by convective mixing during winter has been highlighted as another mechanism that can make iron from deeper water masses available to the surface with fluxes of up to 33.2 μmol Fe m⁻² yr⁻¹ being observed¹⁰. Based on existing Argo float profiles, we tested the hypothesis that the bloom magnitude was linked to the preceding winter maximum ML depth from 2007 to 2019 (see Methods). We found that no significant differences existed in preceding winter (i.e. September) maximum ML depths between years with (mean of 124.9 ± 6.3 m) and years without (mean of 113.1 ± 13.3 m) an open ocean bloom ($p > 0.05$, Supplementary Figure 4e).”

12. Lines 187-190: I don’t understand the logic here because the iron is typically not limiting at the time of bloom initiation (since it has been resupplied through wintertime mixing). Usually we might expect a progression from light limitation early in the season to nutrient limitation later in the season. If this is the case, upwelling shouldn’t necessarily trigger the bloom (although it’s obviously important in sustaining it). von Berg et al. (2020) showed a strong relationship between bloom initiation and the timing of sea ice retreat in this region.

We thank the reviewer for catching the wrong logic here. Indeed, iron should not be limiting at the moment of bloom initiation and relief from light limitation is most probably what triggers the bloom initiation. We have modified this sentence to account for the fact that the upwelling of deep waters is what sustains this large phytoplankton bloom rather than what initiates it:

“When zonal winds peak earlier in the season in February-April as is the case when the open ocean bloom occurs (Supplementary Figure 4c), this induces strong ocean surface stress and provides favorable preconditioning for an upwelling of iron-rich deeper waters, which likely sustains the large phytoplankton bloom through February-March (Supplementary Figure 4a). This also provides an explanation as to why we typically observe the open ocean bloom in February-March, late in the summer production season.”

In addition, we added a reference to the von Berg et al. (2020) study in the sentence of the manuscript where we discuss relief from light limitation following ice retreat.

“By pushing the sea ice south, towards the coast, enhanced easterlies may also relieve phytoplankton production from possible light limitation by the sea ice cover².”

13. Lines 197-198: Yes, but in that study, “upwell” referred to crossing 1000 m, which is not necessarily synonymous with where waters reach the surface ocean (which is what we actually care about for phytoplankton growth).

The reviewer is right and the way this sentence was originally written was misleading. However, we removed this sentence from the manuscript following the re-organisation of this particular section as described in our response to the Reviewer’s comment #15.

14. Lines 218-222: I think this statement is too strong, the isotopic Helium measurements support hydrothermalism as an iron source, but I don’t see how these data show that it is the “main” iron source.

Following this and other comments made by the reviewer below (comments #15 and #17-19), we modified our interpretation and discussion of the iron sources that likely sustained the large phytoplankton bloom we report here (see our response to the reviewer’s comments #15 and #17-19). In this particular paragraph, we modified the following sentence to:

“Two sections of $\delta^3\text{He}$ from the Global Ocean Data Analysis Project (GLODAP-2)⁴⁰ across the Weddell Gyre, around 0° and 30°E (see Methods), and an additional section of $\delta^3\text{He}$ from a recent expedition along 6°E (Transekttokt 2020/21, see Methods) support hydrothermalism as a major iron source for the open ocean bloom we describe here (Figure 3).”

15. Lines 255-256: I don’t see how you can rule out continental shelf-sourced iron. Although the bloom is far from the coast, iron could be advected offshore below the mixed-layer and then brought up to the surface in the bloom region by the same wind-driven upwelling discussed previously.

We agree with the reviewer that we have originally put too much emphasis on the possible role played by deep hydrothermal iron as the “main” iron source that sustains the large phytoplankton bloom we observed. Therefore, we modified our approach.

First, we combined the two sections on hydrothermal iron and other potential iron sources in one shorter section entitled: “Potential iron sources”. In the new section of the manuscript, we discuss the role of hydrothermal iron as well as the likelihood of other known iron sources to have sustained this particular phytoplankton bloom. As a consequence, the following section was modified as:

“Potential iron sources
Our regional analysis of more than two decades of satellite-derived Chl a and climatological sea ice and wind data suggests that the observed bloom is associated with anomalous wind-driven upwelling of iron from deeper waters in this region. Such a mechanism assumes that upstream sources provide the necessary iron that is needed to allow for such high biomass build-up. Hydrothermal iron has recently been shown to be transported thousands

of kilometers from its origin¹³ and be a key driver of primary production of regions that are of critical importance for the oceanic carbon cycle¹⁴. For example, Ardyna et al. (2019; ref. ¹⁵) highlighted the possible role of iron from hydrothermal vents located west of the Southwest Indian Ridge (SWIR, yellow star in Figure 3a) in the unusually large phytoplankton blooms they observed a few hundred kilometers downstream.

The helium isotopic ratio ($\delta^3\text{He} = 100 \times ((^3\text{He}/^4\text{He})_{\text{observed}} / (^3\text{He}/^4\text{He})_{\text{atmospheric}} - 1) [\%]$) is a tracer of primordial helium originating from the Earth's mantle and elevated $\delta^3\text{He}$ ratios (10-12%) are commonly used to detect plumes downstream of hydrothermal vents¹³. Two sections of $\delta^3\text{He}$ from the Global Ocean Data Analysis Project (GLODAP-2) across the Weddell Gyre, around 0° and 30°E (see Methods), and an additional section of $\delta^3\text{He}$ from a recent expedition along 6°E (Transekttokt 2020/21, see Methods) support hydrothermalism as a major iron source for the open ocean bloom we describe here (Figure 3). The 0° and 30°E sections show patches of high $\delta^3\text{He}$ ratios (10-12%) between 500 and 100 m depth in the northern part of the section around 50-55°S (Figure 3b and d), attributed to deep ocean sources in the ACC region. At 30°E, these patches extend all the way to 60°S (Figure 3d), likely indicating the eastward advection of $\delta^3\text{He}$ enhanced sub-surface water masses originating from the hydrothermal vents located west of the SWIR (red arrow in Figure 3a). However, in the predominantly westward coastal flow regime south of 65°S (yellow arrow in Figure 3a), sub-surface $\delta^3\text{He}$ ratios are higher at the Prime Meridian than at 30°E (Figure 3b), indicating an additional source of hydrothermally enriched waters between these two sections that are located upstream and downstream of the bloom region (green patch in Figure 3a). We suggest that this additional source is associated with the northeastern inflow pathway of the Weddell Gyre¹⁶, which brings hydrothermally enriched waters from downstream of the SWIR southward towards the bloom region (brown arrow in Figure 3a), where iron is made available for biological consumption by local upwelling. This is supported by the $\delta^3\text{He}$ section along 6°E where elevated $\delta^3\text{He}$ ratios are found close to and within the surface mixed layer at the northern end of the transect (9.8 and 6.9% at 150 and 75 m depth, respectively, Figure 3c). In this way, hydrothermal vents may act as the long-range source of iron that sustains and explains the persistence of the open ocean bloom that we observed.

While the above analysis points at hydrothermalism as an important source of iron for the observed bloom, microbial iron remineralization, which can provide 5-10 $\mu\text{mol m}^{-2} \text{d}^{-1}$ of dissolved iron¹⁷, may have subsequently supplied a substantial portion of the iron required to maintain the high biomass bloom. As highlighted above, entrainment of WDW into the surface layer by convective mixing during winter is an important source of iron in the Southern Ocean (up to 33.2 $\mu\text{mol Fe m}^{-2} \text{yr}^{-1}$)¹⁰, even though entrainment anomalies are not linked to bloom anomalies in the present study (Supplementary Figure 4e). The bloom was found over deep waters (3,000 m depth), far from the coast (200 km), and not in the lee of any shallow continental shelf or plateau. However, similar to hydrothermal iron, sedimentary iron¹⁸ may have been transported over long distances beneath the mixed layer and brought up to the surface layer by the upwelling mechanism described above.

Considering other known iron sources, atmospheric deposition of iron plays a minor role at these latitudes¹⁹. The bloom was far from typical iceberg drifting routes, with no signs of glacial meltwater in water mass characteristics. In addition, the contribution of glacial meltwater is of minor importance because of the generally low melt rates of the ice shelves in this region²⁰, ruling out the likelihood of glacial-derived iron²¹. Moreover, we detected no signs of sea-ice meltwater in the water column derived from stable oxygen isotopic ratios ($\delta^{18}\text{O}$) (between -0.07 at the surface and 0.4‰ at 1,000 m depth) from samples collected by a CTD-rosette cast at the northern edge of the bloom (see Methods), suggesting that sea ice meltwater is similarly not a major source of iron for this open ocean bloom²².

16. Lines 258-259: satellite ocean color data cannot be retrieved when sea ice is present, so naturally the satellite data will show that the bloom occurred when sea ice was absent. But this does not rule out the possibility of under-ice blooms.

The reviewer is right and this possibility cannot be ruled out, although under-ice blooms in the Southern Ocean are relatively rare and of low amplitude (average of 1.13 mg Chl *a* m^{-3}) as reported by a recent study by Horvat et al. (2022). However, this sentence was misleading in that phytoplankton blooms cannot be observed by satellites if covered by sea ice. Therefore, we removed this sentence.

17. Lines 264-266: Similar to my comment above, sedimentary iron could in theory be advected long distances beneath the mixed-layer and then brought up to the surface in the bloom region by the upwelling in the gyre.

The reviewer is right and we added this possibility in the new version of the manuscript. We refer the reviewer to the new version of this section as described above in our response to comment #15.

18. Lines 271-273: I don't understand the reasoning for ruling out entrainment. As stated in the following line, even if the iron supplied by entrainment is utilized, subsequent iron recycling could potentially help sustain the bloom.

The reviewer is right and entrainment should not be ruled out as a possible source of iron. We modified our original interpretation and now have entrainment listed as one of the possible iron sources that drive or initiate the open ocean bloom we report here. We refer the reviewer to our response to comments #11 and #15.

19. Lines 276-279: As I said at the start, I feel there is too much focus put on the potential hydrothermal origin of the iron. At this point in reading the paper, the main message has been muddled and I think this whole section could be condensed. Or else, it needs to be better clarified what the significance is (to the rest of the paper) that the iron is of hydrothermal origin.

We believe we have answered this particular point in our responses to the above comments #14-15 and #17-18 made by the reviewer. We thank the reviewer for these comments as we believe they significantly strengthened the new version of this manuscript.

20. Lines 281-311: Similar to my point above, the main message of the paper has been lost somewhat at this point in the manuscript. There is a lot of information in this section, but it's not clear to me what purpose it serves to the broader story.

We think that this section conveys important information in that the phytoplankton bloom was overly dominated by one phytoplankton species, *Chaetoceros dichaeta*, which is a large diatom that is known to be an important carbon exporter in this region of the Southern Ocean. Therefore, we would like to keep the section as part of the flow of the manuscript. However, we understand the reviewer's point that the connection between this section and the previous part of the manuscript was not necessarily clear. Therefore, we added the following sentence at the beginning of this section to improve the flow of the manuscript:

"In addition to high biomass, the taxonomical composition of this open ocean bloom has important biogeochemical and ecological consequences."

21. Lines 314-325: Sorry to keep repeating myself, but again, there's tons of information in this paragraph, but I think you need to better establish at the start of the section how this information fits into the broader story. This feels disjointed from the previous section, which in turn, felt disjointed from the section before that.

Similar to the previous comment, we believe that this part of the story is important to discuss to fully describe the outstanding ecosystem feature we observed. However, we understand the reviewer's point that the connection between this section and the previous ones could be improved. Therefore, we added the following sentence at the beginning of this section to improve the flow of the manuscript:

"In addition to the disproportionately important role of the Southern Ocean's biological carbon pump in global climate, phytoplankton blooms also sustain the Southern Ocean's rich food-web, dominated by Antarctic krill, marine birds, seals and both baleen and toothed whales."

22. Lines 342-392: Macrofauna are not my area of specialty, so I can't comment much on the scientific content in section, but I can say that it reads like a completely different paper to me. As above, I think the main message of the paper needs to be better clarified and then this information needs to be better integrated into that message.

We believe we answered this comment in our response to the reviewer's previous comment #21. In addition, we removed some of the text from this section to shorten it and improve the flow of the manuscript. We refer the reviewer to the new version of the manuscript.

23. Lines 395-408: I think the conclusion section will be strengthened once the main message of the paper is better established. Note that although a significant portion of the paper is dedicated to discussing the hydrothermal origin of the iron, it is not mentioned anywhere in this section, which seemingly confirms my impression that it is not as crucial to the story being told here to warrant so much space in the manuscript.

We thank the reviewer for this comment. We have re-worked the Conclusion to better reflect the main message of the paper and to include the possible role of other iron sources (including hydrothermal iron) in sustaining the open ocean bloom. The new version of this Conclusion reads:

"The evidence we report here points towards the importance of this open ocean bloom for the eastern Weddell Gyre ecosystem; particularly the characteristics directly observed at sea (i.e. high phytoplankton carbon biomass and abundant krill), its recurrence during favorable atmospheric and oceanographic conditions (i.e., stronger easterlies, upwelling and less sea ice), the likely additional sources of iron sustaining it (including hydrothermal iron, entrainment and remineralization), and its seemingly long-term significance for the biological carbon pump and top trophic level consumers (marine birds and whales). The timing of the wind and sea ice conditions are important for the open ocean bloom which typically happens in late summer, i.e., most often in February. Such a productive bloom late in the summer is significant for example for krill that need to prepare for the long austral winter²³ or for Antarctic petrels that are raising and fledging chicks in February-March²⁴.

The next legitimate question concerns the fate of this open ocean bloom and its associated ecosystem in the context of climate change. A recent study described major changes in the Southern Ocean physical and biogeochemical properties²⁵ due to poleward intensifying winds and increases in meltwater over the last two decades, which could have large consequences on the fate of this open ocean bloom. Given its seemingly importance for the Weddell Sea, the driving forces and the fate of this key ecosystem feature warrant future studies."

With this modification and the previous ones brought throughout the manuscript, we believe that the flow of the manuscript and the clarity of the message has improved.

24. Figure 1: It seems strange to me to use a cyclical, nonlinear colormap for chlorophyll and the integrated POC points are not visible against the underlying basemap.

Cyclical colormaps are typical for presenting maps of satellite derived Chlorophyll a. We believe this colormap is best to show the contrast that exists between the different areas. However, as Reviewer #1 suggested, we modified the colour scale as the range was not appropriate for the dynamic range of the chlorophyll concentration shown in the figure.

We also added an insert with a zoom in the core area of the bloom to better show the integrated POC markers. We refer the reviewer to the new version of Figure 1a.

25. Change the color of the MLD line (maybe black?) as it's hard to see against the POC colormap and panel c) is not visible to people with red-green colorblindness.

We thank the reviewer for this suggestion, we changed of the MLD to black. We also modified the colour scale of panel c) as suggested by Reviewer #1.

26. Figure 4: Again, this colormap is odd to me for chlorophyll. No legend for the heat map in panel c)

Similar to the Reviewer's comment #24, we believe that cyclical colormaps are typical for presenting maps of satellite derived Chlorophyll a, and are best to show the contrast that exists between the different areas.

However, as Reviewer #1 suggested, we modified the colour scale as the range was not appropriate for the dynamic range of the chlorophyll concentration shown in the figure.

In addition, we added a scale to the heat map in panel c). We also did this for all panels in Figure S6.

References used in the Response letter

1. MacGilchrist GA, Naveira Garabato AC, Brown PJ, Jullion L, Bacon S, Bakker DCE, *et al.* Reframing the carbon cycle of the subpolar Southern Ocean. *Science Advances* 2019, **5**(8): eaav6410.
2. Brown PJ, Jullion L, Landschützer P, Bakker DCE, Naveira Garabato AC, Meredith MP, *et al.* Carbon dynamics of the Weddell Gyre, Southern Ocean. *Global Biogeochemical Cycles* 2015, **29**(3): 288-306.
3. Honjo S. Particle export and the biological pump in the Southern Ocean. *Antarctic Science* 2004, **16**(4): 501-516.
4. Kauko HM, Hattermann T, Ryan-Keogh T, Singh A, de Steur L, Fransson A, *et al.* Phenology and Environmental Control of Phytoplankton Blooms in the Kong Håkon VII Hav in the Southern Ocean. 2021, **8**.
5. Moreau S, Boyd PW, Strutton PG. Remote assessment of the fate of phytoplankton in the Southern Ocean sea-ice zone. *Nature Communications* 2020, **11**(1): 3108.
6. Boyd PW, Arrigo KR, Strzepek R, Van Dijken GL. Mapping phytoplankton iron utilization: Insights into Southern Ocean supply mechanisms. *Journal of Geophysical Research: Oceans* 2012, **117**(6).
7. Arteaga L, Pahlow M, Oschlies A. Modeled Chl:C ratio and derived estimates of phytoplankton carbon biomass and its contribution to total particulate organic carbon in the global surface ocean. *Global Biogeochemical Cycles* 2016, **30**(12): 1791-1810.
8. von Berg L, Prend CJ, Campbell EC, Mazloff MR, Talley LD, Gille ST. Weddell Sea Phytoplankton Blooms Modulated by Sea Ice Variability and Polynya Formation. *Geophysical Research Letters* 2020, **47**(11): e2020GL087954.
9. de Boyer Montégut C, Madec G, Fischer AS, Lazar A, Iudicone D. Mixed layer depth over the global ocean: An examination of profile data and a profile-based climatology. *Journal of Geophysical Research* 2004, **109**: 1-20.
10. Tagliabue A, Sallee J-B, Bowie AR, Levy M, Swart S, Boyd PW. Surface-water iron supplies in the Southern Ocean sustained by deep winter mixing. *Nature Geosci* 2014, **7**(4): 314-320.
11. Martin JH, Fitzwater SE, Gordon RM. Iron deficiency limits phytoplankton growth in Antarctic waters. *Global Biogeochemical Cycles* 1990, **4**(1): 5-12.
12. Klunder MB, Laan P, Middag R, De Baar HJW, van Ooijen JC. Dissolved iron in the Southern Ocean (Atlantic sector). *Deep Sea Research Part II: Topical Studies in Oceanography* 2011, **58**(25): 2678-2694.

13. Resing JA, Sedwick PN, German CR, Jenkins WJ, Moffett JW, Sohst BM, *et al.* Basin-scale transport of hydrothermal dissolved metals across the South Pacific Ocean. *Nature* 2015, **523**(7559): 200-203.
14. Tagliabue A, Resing J. Impact of hydrothermalism on the ocean iron cycle. *Philosophical Transactions of the Royal Society A: Mathematical, Physical and Engineering Sciences* 2016, **374**(2081): 20150291.
15. Ardyna M, Lacour L, Sergi S, d'Ovidio F, Sallée J-B, Rembauville M, *et al.* Hydrothermal vents trigger massive phytoplankton blooms in the Southern Ocean. *Nature Communications* 2019, **10**(1): 2451.
16. Ryan S, Schröder M, Huhn O, Timmermann R. On the warm inflow at the eastern boundary of the Weddell Gyre. *Deep Sea Research Part I: Oceanographic Research Papers* 2016, **107**: 70-81.
17. Boyd PW, Strzepak R, Chiswell S, Chang H, DeBruyn JM, Ellwood M, *et al.* Microbial control of diatom bloom dynamics in the open ocean. *Geophysical Research Letters* 2012, **39**(18): n/a-n/a.
18. Lasbleiz M, Leblanc K, Blain S, Ras J, Cornet-Barthaux V, Hélias Nunige S, *et al.* Pigments, elemental composition (C, N, P, and Si), and stoichiometry of particulate matter in the naturally iron fertilized region of Kerguelen in the Southern Ocean. *Biogeosciences* 2014, **11**(20): 5931-5955.
19. Jickells T, Moore CM. The Importance of Atmospheric Deposition for Ocean Productivity. *Annual Review of Ecology, Evolution, and Systematics* 2015, **46**(1): 481-501.
20. Nøst OA, Biuw M, Tverberg V, Lydersen C, Hattermann T, Zhou Q, *et al.* Eddy overturning of the Antarctic Slope Front controls glacial melting in the Eastern Weddell Sea. *Journal of Geophysical Research: Oceans* 2011, **116**(C11): n/a-n/a.
21. Raiswell R, Tranter M, Benning LG, Siegert M, De'ath R, Huybrechts P, *et al.* Contributions from glacially derived sediment to the global iron (oxyhydr)oxide cycle: implications for iron delivery to the oceans. *Geochim Cosmochim Acta* 2006, **70**.
22. Duprat L, Corkill M, Genovese C, Townsend AT, Moreau S, Meiners KM, *et al.* Nutrient distribution in East Antarctic summer sea ice: a potential iron contribution from glacial basal melt. *Journal of Geophysical Research: Oceans* 2020, **n/a**(n/a): e2020JC016130.
23. Meyer B. The overwintering of Antarctic krill, *Euphausia superba*, from an ecophysiological perspective. *Polar Biology* 2012, **35**(1): 15-37.
24. Fauchald P, Tarroux A, Tveraa T, Cherel Y, Ropert-Coudert Y, Kato A, *et al.* Spring phenology shapes the spatial foraging behavior of Antarctic petrels. *Marine Ecology Progress Series* 2017, **568**: 203-215.
25. Bronselaer B, Russell JL, Winton M, Williams NL, Key RM, Dunne JP, *et al.* Importance of wind and meltwater for observed chemical and physical changes in the Southern Ocean. *Nature Geoscience* 2020, **13**(1): 35-42.

REVIEWERS' COMMENTS

Reviewer #1 (Remarks to the Author):

Second review of “Wind-driven upwelling of iron sustains dense phytoplankton blooms and productive food webs in the eastern Weddell Gyre” by Moreau et al.

General comments: This is my second review of this manuscript. Same as before, I think this manuscript represents novel and important ecological research for the Antarctic region. The most notable strength of this study is the ample collection of datasets employed to describe the bloom, discuss its potential fueling by hydrothermal sources, and its ecological significance. The authors have addressed most of my comments in their rebuttal letter and through revisions in the manuscript. I still have a few specific observations that I detail below.

- Figure 1: I think panel (a) looks much better. However, I have a printed copy of the earlier manuscript and I think panels (c) and (d) looked better before. Perhaps the log conversion was not helpful here as the concentrations of these properties are so high over the entire mixed layer that it results in a gaussian and not logarithmic distribution. (contrary to what is usually seen for phytoplankton). I encourage the authors to compare both versions and choose whatever yields the highest contrast, specifically regarding the deep chlorophyll maximum around 68.3 S, which is still hard (perhaps harder than before) to see. Also, I think a multi-color colormap (instead of a monochromatic one) would help here (similar to what is used for the helium profiles in Figure 3). Consider this suggestion also for Figure S1.

- Abstract: Line 36: Consider replacing “fueled” by “likely driven” or something similar. The wind anomalies are not fueling the bloom. The bloom is fueled by the nutrients upwelled by the wind.

- Line 53: I made this point before, and I think it was not properly addressed. It is still not clear if the 50 Tg C/yr are exported below a depth horizon comparable to that reported in Honjo et al. (2008). Different depth horizon can render different export efficiency patterns (see Palevsky and Doney, 2021, <https://agupubs.onlinelibrary.wiley.com/doi/full/10.1029/2020GB006790>).

- Line 174. Is $r = -0.23$ the average over the area? Please clarify. You should report the higher statistic for the 1-month lag correlation.

- Line 189: “*When zonal winds peak earlier in the season in February-April as is the case when the open ocean bloom occurs (Supplementary Figure 4c), this induces strong ...*” I suggest rewording this phrase.

- Line 247: “*As highlighted above, entrainment of WDW into the surface layer by convective mixing during winter is an important source of iron in the Southern Ocean (up to $33.2 \mu\text{mol Fe m}^{-2} \text{yr}^{-1}$)²⁸, even though entrainment anomalies are not linked to bloom anomalies in the present study (Supplementary Figure 4e).*” This seems contradicting your argument for hydrothermal sources being the main source of iron for the bloom. I am not surprised that the correlation between entrainment and biomass anomalies is not large. Deep entrainment would reduce iron limitation only when the entrained deep water has a large iron concentration, which does not need to be constant nor correlate with periods of deepest entrainment. I think both mechanisms

(i.e., entrainment and strong transport of deep waters with high iron concentration) likely complement each other to produce the largest blooms.

- Line 595: *"The analysis of satellite-derived ocean color was complemented with using SeaWiFs"*
Remove "with".

Reviewer #2 (Remarks to the Author):

Issues have been addressed properly.

However, one section of text is still unclear:

Ln 128: dFe was elevated and probably not limiting throughout the study area. However, before and afterwards it is stated that Fv/Fm was low in the bloom, suggesting that iron was limiting in the bloom. I would suggest to make the distinction between the bloom area and the general study area more clear here.

Reviewer #3 (Remarks to the Author):

Thanks to the authors for considering and incorporating my feedback. From my perspective, the flow and message of the paper is much stronger in the revised manuscript. I think this is a useful contribution to the field and I recommend publication.

We answer below each point raised by the three reviewers. Our answers are written in blue. Text from the manuscript is copied in italic and the modified parts of the manuscript are in italic and underlined.

REVIEWERS' COMMENTS

Reviewer #1 (Remarks to the Author):

Second review of “Wind-driven upwelling of iron sustains dense phytoplankton blooms and productive food webs in the eastern Weddell Gyre” by Moreau et al.

General comments: This is my second review of this manuscript. Same as before, I think this manuscript represents novel and important ecological research for the Antarctic region. The most notable strength of this study is the ample collection of datasets employed to describe the bloom, discuss its potential fueling by hydrothermal sources, and its ecological significance. The authors have addressed most of my comments in their rebuttal letter and through revisions in the manuscript. I still have a few specific observations that I detail below.

We thank the reviewer for the very positive comments on our manuscript. We answer below each point raised by the reviewer.

- Figure 1: I think panel (a) looks much better. However, I have a printed copy of the earlier manuscript and I think panels (c) and (d) looked better before. Perhaps the log conversion was not helpful here as the concentrations of these properties are so high over the entire mixed layer that it results in a gaussian and not logarithmic distribution. (contrary to what is usually seen for phytoplankton). I encourage the authors to compare both versions and choose whatever yields the highest contrast, specifically regarding the deep chlorophyll maximum around 68.3 S, which is still hard (perhaps harder than before) to see. Also, I think a multi-color colormap (instead of a monochromatic one) would help here (similar to what is used for the helium profiles in Figure 3). Consider this suggestion also for Figure S1.

We thank the reviewer for revising these 2 figures and providing advice on how to best represent these features.

We tested which colour scale (monochromatic versus polychromatic and normal versus log scales) best represent these features, including the representation of the deep Chlorophyll *a* maximum (Figure 1c-d) and the export of particles and phytodetritus (Figure S1).

We find that the normal scale is best to represent the distribution of Chlorophyll *a* and POC in the upper 60 m, particularly the deep Chlorophyll *a* maximum. In addition, we increased the resolution of the colour scales in Figure 1c-d. However, we kept the monochromatic colour scales in these two panels to differentiate them from the other figures in the manuscript and, thus, avoid any confusion for the reader.

In addition, we also find that the normal scale best represents the export of particles and phytodetritus in Supplementary Figure 1. We also kept the monochromatic colour scales in these two panels to differentiate them from the other figures in the manuscript and be consistent with the representation of Chlorophyll *a* and POC in Figure 1.

With these changes, we think that Figures 1 and S1 now best represent these features.

- Abstract: Line 36: Consider replacing “fueled” by “likely driven” or something similar. The wind anomalies are not fueling the bloom. The bloom is fueled by the nutrients upwelled by the wind.

We thank the reviewer for this comment. We modified the sentence as suggested. The new version of the sentence reads:

“We show that, over 1997-2019, this open ocean bloom was likely driven by anomalies in easterly winds that push sea ice southwards and favor the upwelling of Warm Deep Water enriched in hydrothermal iron and, possibly, other iron sources.”

- Line 53: I made this point before, and I think it was not properly addressed. It is still not clear if the 50 Tg C/yr are exported below a depth horizon comparable to that reported in Honjo et al. (2008). Different depth horizon can render different export efficiency patterns (see Palevsky and Doney, 2021, <https://agupubs.onlinelibrary.wiley.com/doi/full/10.1029/2020GB006790>).

We understand the reviewer’s point and agree that this point was not fully clear in the previous version of the manuscript. This statement comes from the study of MacGillchrist et al. (2019, ref. [1]) where they estimate that Deep Weddell Sea Water that comes out of the Gyre carries a significant carbon enrichment with it, which comes from the remineralization of the exported organic matter within the Gyre, before it descends underneath the AAC following the northward limb of the global overturning circulation. This process acts to draw carbon into the World Ocean abyss as described by MacGillchrist et al. (2019, ref. [1]). This conclusion by the authors relies on previous findings by Garrabato et al. (2002, ref. [2]) that Weddell Deep Water leaves the Weddell Gyre at depth between 1,500 m and 4,000 m depth. In conclusion, a simple answer to the reviewer’s comment is that the 50 Tg C/yr exported below the ACC and reported by MacGillchrist et al. (2019, ref. [1]) is, indeed, significant when compared to the global ocean biological carbon pump estimated at 430 Tg C yr⁻¹ across the 2,000 m horizon by Honjo et al. (2008, ref. [3]).

In the new version of the manuscript, we modified this sentence to further clarify this point:

“A recent carbon budget across the Weddell Gyre^[1], based on observations of inorganic carbon, suggests that intense primary productivity in the region of the Weddell Gyre, away from continental shelves, can explain the carbon drawdown estimated from the carbon deficit directed out of the Gyre below the ACC (i.e., at depth between 1,500 and 4,000 m)^[2, 4]. The strength of the resulting Weddell Gyre carbon dioxide (CO₂) sink to depth, at an estimated ~50 Tg C yr⁻¹, is significant when compared to the global ocean biological carbon pump, estimated at 430 Tg C yr⁻¹ across the 2,000 m horizon by Honjo et al. (2008; ref. ^[3]).”

- Line 174. Is $r = -0.23$ the average over the area? Please clarify. You should report the higher statistic for the 1-month lag correlation.

The value $r = -0.23$ corresponds to the r -value at the location of the bloom (shown with a black cross in Supplementary Figure 5a-b). We followed the reviewer suggestion and now report the higher statistic for the 1-month lag correlation. The sentence was modified as:

*“A weak but significant negative correlation exists between the satellite-derived Chl *a* de-seasoned anomaly and the zonal wind de-seasoned anomaly in the area of the open ocean bloom (with a maximum r -value of -0.32; see Methods and Supplementary Figure 5a-b).”*

- Line 189: *“When zonal winds peak earlier in the season in February-April as is the case when the open ocean bloom occurs (Supplementary Figure 4c), this induces strong ...”* I suggest rewording this phrase.

We modified and split the sentence in two sentences:

“However, when an open ocean bloom occurs, the zonal winds peak earlier in the season between February-April (Supplementary Figure 4c). This induces strong ocean surface stress and provides favorable preconditioning for an upwelling of iron-rich WDW, which likely sustains the large phytoplankton bloom through February-March (Supplementary Figure 4a).”

- Line 247: “As highlighted above, entrainment of WDW into the surface layer by convective mixing during winter is an important source of iron in the Southern Ocean (up to $33.2 \mu\text{mol Fe m}^{-2} \text{ yr}^{-1}$)²⁸, even though entrainment anomalies are not linked to bloom anomalies in the present study (Supplementary Figure 4e).” This seems contradicting your argument for hydrothermal sources being the main source of iron for the bloom. I am not surprised that the correlation between entrainment and biomass anomalies is not large. Deep entrainment would reduce iron limitation only when the entrained deep water has a large iron concentration, which does not need to be constant nor correlate with periods of deepest entrainment. I think both mechanisms (i.e., entrainment and strong transport of deep waters with high iron concentration) likely complement each other to produce the largest blooms.

We agree with the reviewer that deep winter entrainment and wind-driven upwelling would act on the same water mass to bring deep iron rich water to the surface, complementing each other. However, we think that entrainment of WDW into the surface layer by convective mixing during winter and wind-driven upwelling of hydrothermal iron act on different time scale in the area of the bloom we describe. Entrainment by convective mixing during winter would only resupply deep hydrothermal iron into the surface layer before the beginning of the production season. On the contrary, wind-driven upwelling of hydrothermal iron may continuously resupply the surface water with deep iron while the process is active during the productive season, i.e. between February-April when zonal winds peak when an open ocean bloom occurs (Supplementary Figure 4c). This may explain the lack of correlation between entrainment and bloom anomalies.

We modified the following sentence to clarify this point:

“As highlighted above, entrainment of WDW into the surface layer by convective mixing during winter is an important source of iron before the start of the productive season in the Southern Ocean (up to $33.2 \mu\text{mol Fe m}^{-2} \text{ yr}^{-1}$)⁽⁶⁾, even though entrainment anomalies are not linked to bloom anomalies in the present study (Supplementary Figure 4e).”

- Line 595: “The analysis of satellite-derived ocean color was complemented with using SeaWiFs”
Remove “with”.

We modified the sentence as suggested by the reviewer.

Reviewer #2 (Remarks to the Author):

Issues have been addressed properly.

However, one section of text is still unclear:

Ln 128: dFe was elevated and probably not limiting throughout the study area. However, before and afterwards it is stated that Fv/Fm was low in the bloom, suggesting that iron was limiting in the bloom. I would suggest to make the distinction between the bloom area and the general study area more clear here.

We thank the reviewer for the positive comments on our manuscript. We modified the following sentences to improve the distinction between the bloom area and the larger study area:

“During our campaign, dissolved iron (dFe) was relatively elevated[7], and probably not limiting, at the sea surface throughout the larger study area, ranging from $0.65 \pm 0.2 \text{ nM}$ at Astrid Ridge, $0.4 \pm 0.1 \text{ nM}$ at Maud Rise and $0.6 \pm 0.04 \text{ nM}$ at 2 stations studied south of the bloom we report here (Supplementary Figure 2b). Unfortunately, dissolved iron concentrations were not measured directly inside the open ocean bloom area during the campaign and these concentrations should only be considered representative of the larger study area presented in Kauko et al. (2021, ref. [8]). The accumulation of both Chl a and POC at the base of the mixed layer and the very low quantum yield of photosystem II, F_v/F_m may, however, indicate that iron was becoming limiting at the time of sampling.”

Reviewer #3 (Remarks to the Author):

Thanks to the authors for considering and incorporating my feedback. From my perspective, the flow and message of the paper is much stronger in the revised manuscript. I think this is a useful contribution to the field and I recommend publication.

We thank the reviewer for the very positive comments on our manuscript.

References

1. MacGilchrist, G.A., et al., *Reframing the carbon cycle of the subpolar Southern Ocean*. Science Advances, 2019. **5**(8): p. eaav6410.
2. Garabato, A.C.N., et al., *On the export of Antarctic Bottom Water from the Weddell Sea*. Deep Sea Research (Part II, Topical Studies in Oceanography), 2002. **49**: p. 4715-4742.
3. Wollenburg, J.E., et al., *Ballasting by cryogenic gypsum enhances carbon export in a Phaeocystis under-ice bloom*. Scientific Reports, 2018. **8**(1): p. 7703.
4. Brown, P.J., et al., *Carbon dynamics of the Weddell Gyre, Southern Ocean*. Global Biogeochemical Cycles, 2015. **29**(3): p. 288-306.
5. Honjo, S., *Particle export and the biological pump in the Southern Ocean*. Antarctic Science, 2004. **16**(4): p. 501-516.
6. Tagliabue, A., et al., *Surface-water iron supplies in the Southern Ocean sustained by deep winter mixing*. Nature Geosci, 2014. **7**(4): p. 314-320.
7. Klunder, M.B., et al., *Dissolved iron in the Southern Ocean (Atlantic sector)*. Deep Sea Research Part II: Topical Studies in Oceanography, 2011. **58**(25): p. 2678-2694.
8. Kauko, H.M., et al., *Phenology and Environmental Control of Phytoplankton Blooms in the Kong Håkon VII Hav in the Southern Ocean*. 2021. **8**.